# An On-Demand Drug Delivery System for Control of Epileptiform Seizures

**DOI:** 10.3390/pharmaceutics14020468

**Published:** 2022-02-21

**Authors:** Takashi Nakano, Shakila B. Rizwan, David M. A. Myint, Jason Gray, Sean M. Mackay, Paul Harris, Christopher G. Perk, Brian I. Hyland, Ruth Empson, Eng Wui Tan, Keshav M. Dani, John NJ Reynolds, Jeffery R. Wickens

**Affiliations:** 1Neurobiology Research Unit, Okinawa Institute of Science and Technology Graduate University, Okinawa 904-0495, Japan; nakano.takashi@gmail.com; 2School of Pharmacy, University of Otago, Dunedin 9016, New Zealand; shakila.rizwan@otago.ac.nz; 3Department of Chemistry, University of Otago, Dunedin 9016, New Zealand; dmyint@atascientific.com.au (D.M.A.M.); sean.mackay@otago.ac.nz (S.M.M.); ewtan@chemistry.otago.ac.nz (E.W.T.); 4Department of Anatomy, University of Otago, Dunedin 9016, New Zealand; jason.gray@otago.ac.nz (J.G.); cperk@umass.edu (C.G.P.); john.reynolds@otago.ac.nz (J.N.R.); 5Callaghan Innovation, Wellington 5010, New Zealand; paul.harris@callaghaninnovation.govt.nz; 6Department of Physiology, University of Otago, Dunedin 9016, New Zealand; brian.hyland@otago.ac.nz (B.I.H.); ruth.empson@otago.ac.nz (R.E.); 7Femtosecond Spectroscopy Unit, Okinawa Institute of Science and Technology Graduate University, Okinawa 904-0495, Japan; kmdani@oist.jp

**Keywords:** liposome, nanoparticle, seizure, laser, ultrasound

## Abstract

Drug delivery systems have the potential to deliver high concentrations of drug to target areas on demand, while elsewhere and at other times encapsulating the drug, to limit unwanted actions. Here we show proof of concept *in vivo* and *ex vivo* tests of a novel drug delivery system based on hollow-gold nanoparticles tethered to liposomes (HGN-liposomes), which become transiently permeable when activated by optical or acoustic stimulation. We show that laser or ultrasound simulation of HGN-liposomes loaded with the GABA_A_ receptor agonist, muscimol, triggers rapid and repeatable release in a sufficient concentration to inhibit neurons and suppress seizure activity. In particular, laser-stimulated release of muscimol from previously injected HGN-liposomes caused subsecond hyperpolarizations of the membrane potential of hippocampal pyramidal neurons, measured by whole cell intracellular recordings with patch electrodes. In hippocampal slices and hippocampal–entorhinal cortical wedges, seizure activity was immediately suppressed by muscimol release from HGN-liposomes triggered by laser or ultrasound pulses. After intravenous injection of HGN-liposomes in whole anesthetized rats, ultrasound stimulation applied to the brain through the dura attenuated the seizure activity induced by pentylenetetrazol. Ultrasound alone, or HGN-liposomes without ultrasound stimulation, had no effect. Intracerebrally-injected HGN-liposomes containing kainic acid retained their contents for at least one week, without damage to surrounding tissue. Thus, we demonstrate the feasibility of precise temporal control over exposure of neurons to the drug, potentially enabling therapeutic effects without continuous exposure. For future application, studies on the pharmacokinetics, pharmacodynamics, and toxicity of HGN-liposomes and their constituents, together with improved methods of targeting, are needed, to determine the utility and safety of the technology in humans.

## 1. Introduction

Conventional drug treatments aim to minimize the side-effects of drugs by targeting specific receptors or bodily compartments. For diseases with episodic, paroxysmal expression, another way to limit side-effects is to reduce exposure to active drugs inside the body by delivering the drug only when and where it is needed [1]. Time-selective focal triggering of drug release has the potential to reduce side-effects by limiting drug exposure to the specific time period and location at which therapeutic effects occur [2]. Such a possibility requires the development of on-demand drug delivery devices that sequester drugs so that they remain inert until required, and trigger release when necessary. We here report the electrophysiological effects of on-demand release of the gama aminobuturic acid (GABA) agonist, muscimol, from hollow gold nanoparticle-tethered liposomes (HGN-liposomes) and its effectiveness in attenuating seizures in experimental animal models.

Liposomes are phospholipid-based vesicles composed of a lipid bilayer, in which a wide range of drugs can be encapsulated [3]. Drug release from liposomes can be triggered by external stimulation, such as hyperthermia [4]. Earlier studies aimed at discharging the entire contents of the liposomes in a single release event [5]. Later studies investigated the possibility of repeated release from liposomes over an extended life-time in the body. Our recently developed HGN-liposome system provides repetitive, on-demand release *ex vivo*, with the temporal profile and quantity of release controlled by varying laser power and exposure duration or pulses of low-intensity, therapeutic ultrasound (US) [6]. Here, we apply these technologies to release muscimol from liposomes within the extracellular matrix of the mammalian brain.

We used muscimol (3-hydroxy-5-aminomethylisoxazole) in the present study because it is a potent and selective GABA_A_ receptor agonist, used extensively in electrophysiological studies of GABAergic inhibitory neurotransmission. Muscimol potently and reversibly inhibits neuronal activity, and thus has been considered to have the potential to suppress an epileptic seizure [7]. The metabolism of muscimol in both the brain and periphery is largely through the removal of an amino group by transamination [8]. In the mouse, about 1/3 is excreted as muscimol, 1/3 as a cationic conjugate, and 1/3 as an oxidation product [9]. The rapid clearance of muscimol in the periphery, and its slow passage across the blood–brain barrier (BBB), mean that high doses are needed when given intravenously, causing adverse effects, and making it unsuitable for systemic use in the treatment of epilepsy [10,11].

In contrast to systemic administration, when delivered transmeningeally in experimental animals, muscimol has antiepileptic effects, without the adverse effects associated with systemic delivery [12,13,14,15]. Direct injection of muscimol into the brain is orders-of-magnitude more effective than intravenous injection. For example, nanomomolar concentrations injected into brain produce similar effects to micromolar concentrations injected intravenously [16,17]. When injected locally into the brain in low μg quantities, muscimol produced no sedation or other central side-effects [18,19]. Similarly, studies of convection enhanced delivery of muscimol into the brain of non-human primates and patients with drug-resistant epilepsy, as well as other disorders, have shown that it is safe, with no adverse effects [20,21,22]. Thus, muscimol is a potential anti-epileptic treatment with few side-effects, provided it can be delivered directly to the brain. Muscimol is, therefore, a suitable candidate for proof of concept of the HGN-liposome delivery system.

Here, we used muscimol-loaded HGN-liposomes to produce repetitive on-demand release of muscimol within the extracellular matrix of the brain. We aimed to determine whether release of muscimol from HGN-liposomes by laser or ultrasound stimulation in live brain tissue was effective in attenuating seizure activity. We used three well-known models of seizure activity: two *ex vivo* models that rely on removal of Mg^2+^ ions and repetitive stimulation [23,24,25]; and *in vivo* pentylenetetrazol (PTZ), to cause seizures that propagate to status epilepticus [26]. We found that when muscimol-containing HGN-liposomes were present in brain slices, laser or US stimulation released muscimol on-demand and caused neural inhibition and arrest of seizure activity. Similarly, in whole animals, we found that US stimulation of the brain after intravenous injection of muscimol loaded HGN-liposomes was effective in reducing seizure activity.

## 2. Methods

### 2.1. Animals

A total of 11 mice and 58 rats were used in the research. Animals were handled in accordance with protocols approved by the Okinawa Institute of Science and Technology Animal Care and Use Committee (*ex vivo* hippocampal seizure model) and the University of Otago Animal Ethics Committee (*ex vivo* entorhinal cortex seizure model, and *in vivo* seizure model). In the *ex vivo* hippocampal experiments, brain slices were obtained from *n* = 6 male, 3 to 8-week-old Swiss Webster mice. Mice were group housed with littermates on reversed light cycle, with free access to standard chow and water. An additional *n* = 5 male 3 to 5-week-old Swiss Webster mice were used to test liposome ability to sequester contents in absence of stimulation. After injection, these mice were individually housed until perfused for histology. In the *ex vivo* entorhinal cortex experiments, brain slices were obtained from 40 male and female 4 to 8-week-old Wistar rats. In the *in vivo* experiments a total of *n* = 18 male Wistar rats were used, group housed in standard open top cages under reverse light cycle, and fed standard rat chow and water *ad libitum*. The targeted weight range was 250 to 300 g. These were allocated to four groups, unbiased by any animal-related factors (PTZ-only, *n* = 3; PTZ plus HGN-liposome, *n* = 3; US without liposomes, *n* = 4; and PTZ plus HGN-liposome plus US, *n* = 8).

### 2.2. HGN-Liposome Preparation

Liposomes and hollow gold nanoshells were prepared and assembled, as previously described [6,27,28], by conjugation using a terminal thiol-derived phospholipid (DSPE-PEG2000-SH) to produce a biocompatible drug delivery system that could encapsulate muscimol, a GABA agonist. DSPE-PEG2000-SH was synthesized by combining 1,2-distearoyl-sn-glycero-3-phosphoethanolamine-N-[amino(polyethylene glycol)-2000] (ammonium salt) (DSPE-PEG2000-NH_2_) (100 mg; 35.8 μmol with 2-iminothiolane (10 mg, 73 μmol) in phosphate buffer (3 mmol L^−1^; pH 9.5; 15 mL) for 30 min at room temperature. Sodium chloride (approx. 1 g) was dissolved in the reaction mixture, and the product was subsequently extracted into chloroform and dried over magnesium sulfate. The solvent was then removed by rotary evaporation, and the product was further dried under vacuum for 5 h.

Hollow gold nanoparticles were synthesized by the galvanic replacement of a silver nanoparticle template, as previously reported, resulting in a hydrodynamic diameter of approximately 30 nm and strong absorption in the visible to near-infrared region [29]. HGN-liposomes were prepared using a phospholipid composition previously described by our group [6,27,28]. Chloroform solutions of 1.2-distearoyl-sn-glycero-3-phosphcholine (DSPC), cholesterol, sphingomyelin, 1,2-distearoyl-sn-glycero-3-phosphoethanolamine-N-[methoxy(polyethylene glycol)-2000] (DSPE-PEG2000), DSPE-PEG2000-SH were combined in a molar ratio of 100:5:5:4:3.5. The solvent was removed under vacuum to form a thin lipid film, which was rehydrated using a phosphate-buffered solution (20 mmol L^−1^ Na_2_HPO_4_; pH 5.5) containing either 100 mmol L^−1^ muscimol or 25 mM kainic acid. The lipid suspension was then extruded through 400-nm polycarbonate membranes, to maximize the passive encapsulation of muscimol, producing uniformly sized liposomes for laser studies. However, as liposomes of smaller sizes are generally regarded as more suitable for intravenous administration, liposomes of 200 nm were prepared for ultrasound studies. The suspension of concentrated HGNs (Au concentration 6–10 mg mL^−1^ by inductively coupled plasma mass spectrometry) was added incrementally to the liposome suspension, until a final HGN:liposome ratio of approximately 1:1 was reached (as determined by transmission electron microscopy (Figure 1A). Approximately 200 μL of HGNs with an Au concentration of 7 mg mL^−1^ to 1 mL of 200 nm liposomes, with a total lipid concentration of 10 mmol L^−1^, or approximately ¼ the amount of gold was added to 400 nm liposomes, with an equivalent 10 mmol L^−1^ phospholipid concentration. The HGN-liposome suspension was subsequently dialyzed against phosphate buffered saline (100 mmol L^−1^ NaCl; 20 mmol L^−1^ Na_2_HPO_4_; pH 7.4; 2 L) for 24 h to remove the unencapsulated muscimol.

We have previously reported on release measurements of dopamine, as well as carboxyfluorescein, from ultrasound and laser activated HGN liposomes [6,27]. In the present study, we were unable to do in vitro release studies for muscimol, because we were unable to identify a chemical, electrochemical, or spectroscopic assay that could distinguish between encapsulated and non-encapsulated muscimol and that was sensitive enough to measure its release in real time. Hence, we used the biological assays described below. Further details of the preparation and analysis of liposomal nanostructures and hollow gold nanoshells are given in Appendix A.

### 2.3. Ex Vivo Hippocampal Seizure Model

Mice were deeply anesthetized with isoflurane and decapitated, and the brain was rapidly removed. Horizontal slices, 300 mm thick, containing the hippocampus were cut on a vibratome (VT1200S, Leica Microsystems, Wetzlar, Germany) in cold cutting solution containing the following (in mM): 92.0 N-methyl-D-glucamine (NMDG), 2.5 KCl, 10.0 MgCl_2_, 0.5 CaCl_2_, 1.25 NaH_2_PO_4_, 30.0 NaHCO_3_, 20.0 HEPES, 2.0 thiourea, 5.0 sodium ascorbate, 3.0 sodium pyruvate, and 25.0 glucose, and saturated with 95% O_2_—5% CO_2_, Slices were then incubated in oxygenated artificial cerebrospinal fluid (ACSF) maintained at a temperature of 36 °C for 1 h. The standard ACSF had the following composition (mM): 118.0 NaCl, 2.5 KCl, 2.0 CaCl_2_, 1.0 MgCl_2_, 26.0 NaHCO_3_, 1.25 NaH_2_PO_4_, 1.5 myo-inositol, 0.5 sodium ascorbate, 2.0 sodium pyruvate, and 10.0 Glucose. The composition of low Mg^2+^/high K^+^ ACSF was the same, except for (mM) 5.0 K^+^ and 0.5 Mg^2+^.

After incubation, a single slice was transferred to a recording chamber placed on the stage of an upright microscope, and perfused (3–4 mL/min) with oxygenated ACSF at 32 °C. HGN-liposomes were injected directly into the slice in the region of interest. The remaining slices were kept in a holding chamber containing oxygenated ACSF at room temperature until required.

The experimental setup for the hippocampal slice experiments is shown in Figure 1A. Whole-cell recordings were made from CA1 pyramidal neurons using patch pipettes (4–6 MΩ) filled with internal solution containing the following (mM): 132.0 K gluconate, 6.0 KCL, 6 NaCl_2_, 10.0 HEPES, 2 MgCL_2_, 2.0 NaATP, 0.4 NaGTP, 0.5 EGTA; pH 7.2–7.4. Local field potentials (LFPs) were recorded in the same location using extracellular electrodes positioned in the CA1 stratum pyramidal layer of the subiculum. LFPs were measured using borosilicate glass pipettes (1–2 MΩ) filled with ACSF. Signals were amplified by MultiClamp 700B (Molecular Devices, Union City, CA, USA), digitized at 10,000 Hz, and band-pass filtered over 1–2000 Hz by pCLAMP 10 (Molecular Devices, Silicon Valley, CA, USA). Offline analysis was conducted using MATLAB (MathWorks, Natick, MA, USA).

Optical stimulation of liposomes was delivered using infrared (890 nm) femtosecond (fs) pulsed laser of a 2-photon microscope. Pulse duration was 100 fs, and repetition rate was 80 MHz. Laser pulses were transmitted through a 60× objective lens and made a 430-nm diameter spot in the brain slice. The light source (MaiTai, Coherent, Santa Clara, CA, USA) provided continuous laser power at the source of approximately 2 W, which was attenuated by an acousto-optic modulator. The laser stimulation was set using software (FluoView, Olympus, Tokyo, Japan) to a scan area of 211.14 µm × 211.14 µm and a sampling speed of 10 µs/pixel.

In the hippocampal slice model system, seizures were induced by perfusion with low Mg^2+^/high K^+^ ACSF. Spontaneous seizure-like events (SLEs) seldom occurred in response to this treatment alone. When SLEs did not occur spontaneously they were induced by repetitive electrical stimulation. Electrical stimulation (600–1200 µA, 100 µs, monophasic) was applied through a bipolar stimulating electrode placed in CA3, in order to stimulate Schaffer collaterals. The intensity of the stimulation for each slice was adjusted to a value that evoked SLEs in the CA1 area. After initial adjustment, the stimulation intensity remained fixed. To test the effect of muscimol release from liposomes, HGN-liposomes containing muscimol (100 mM, Tocris Bioscience, Tokyo, Japan), were pressure-injected directly into slices via a glass micropipette (tip diameter 50–100 µm) over a period of 1 s.

### 2.4. Ex Vivo Entorhinal Cortex Seizure Model

Experiments using an entorhinal cortex (EC) seizure model were performed on brain slice wedges obtained from 40 male and female 1–2 month-old Wistar rats. Electrophysiological recordings in the EC were made using methods described previously [23]. Briefly, horizontal combined hippocampal EC slices (500 μm thick) were cut using a Vibroslice (Campden Instruments, Leicester, UK). From these slices, a wedge-shaped segment of the EC tissue, 2–3 mm wide, was dissected and transferred to a custom designed two-compartment grease gap chamber (see experimental setup in Figure 1B) continuously perfused with ACSF at room temperature (~1.5 mL/min). The ACSF contained (in mM): NaCl 135, KCl 3, NaH_2_PO_4_ 1.25, MgCl_2_ 2, CaCl_2_ 2, glucose 10, and NaHCO_3_ 26 (all from Sigma, NZ), saturated with 95% O_2_/5% CO_2_. For Mg^2+^-free ACSF, the MgCl_2_ was omitted. An HGN-liposome reservoir was plumbed in and out of the perfusion flow and an ultrasound probe was positioned at the base of the HGN-liposome reservoir to trigger release of muscimol.

Differential recordings were made across the grease gap using Ag/AgCl pellet electrodes (Harvard Apparatus, Waterbeach, UK) located in both chambers (Figure 1B) with an NL102 amplifier (Neurolog, Welwyn Garden City, UK) (×100 gain, high-pass filter with 8.9 Hz cut-off) and a chart recorder (Semat, London, UK). After placing the slice on the grease gap, the slice was perfused with ACSF for 30 min, and thereafter with ACSF lacking Mg^2+^. Removal of Mg^2+^ from the ACSF led to the appearance of repetitive SLEs 40 to 120 min after the switch. A solution of muscimol-containing HGN-liposomes was applied for 15 min, after which the ultrasound (US) trigger was applied to the liposome reservoir (Figure 1B). Slices were perfused with this ultrasonicated liposomal formulation for 30 min before being washed-out with Mg^2+^-free ACSF and the frequency of seizure-like events (SLEs) and late recurrent discharges (LRDs) was measured before and during drug application.

### 2.5. In Vivo Seizure Model

Rats were anesthetized (urethane, 1500 to 1800 mg/kg) and mounted in a stereotaxic frame (Figure 1C). Craniotomies were made on the superior surface of frontal bone (2.7 mm in diameter) and on the lateral side caudal to the left orbit (4.0 mm in diameter). A 4-mm collimator was inserted through the lateral craniotomy and pressed gently against the dura over the lateral aspect of the left frontal lobe, with a layer of acoustic coupling gel between. A silver wire epidural electroencephalogram (EEG) recording electrode was secured using dental cement in the uppermost craniotomy. Epileptiform EEG was induced using 60 mg/kg PTZ administered intravenously via a cannula in the left jugular vein. Animals were divided into four groups (PTZ-only, *n* = 3; PTZ plus HGN-liposome, *n* = 3; US without liposomes, *n* = 4; and PTZ plus HGN-liposome plus US, *n* = 8). In groups exposed to muscimol-loaded HGN-liposomes (90 mM), the formulation was administered intravenously via the jugular cannula and allowed to circulate for 5 min, after which ultrasound was delivered in bursts of 30 sec at 30% duty cycle at 1 MHz. No more than three applications of ultrasound occurred in any given 5 min recording interval.

For analysis, individual power spectra were constructed and normalized, such that the total area under the curve of each plot was 1. The fraction of total area in the EEG frequency bands 0–1 Hz, 1–3 Hz, and 3–5 Hz was calculated. For each rat mean post-PTZ values for the intervals 10–19, 20–29, 30–39, and 40–49 min were derived by averaging the values for the 1st and 2nd 5-min period in each interval. These averaged time epoch data for each rat were then normalized to the values for the 5-min period following PTZ application.

### 2.6. Test of Liposome Ability to Sequester Contents in Absence of Stimulation

For this control experiment, designed to test whether a drug will remain in HGN-liposomes until released by an external trigger, HGN-liposomes containing kainic acid (KA) were injected into the primary somatosensory cortex of mice at stereotaxic coordinates (AP: −1.0 mm, ML: +/−1.5 mm, DV: 1.5 mm) in a volume of 1.0 µL. Positive control injections of KA directly into brain tissue were made in other mice, using a volume of 1 µL in a concentration of 10 nM. One week after the injection of HGN-liposomes, animals were perfused with 4% paraformaldehyde and were brains extracted and post-fixed in the same fixative. Coronal sections (80 µm) using a vibratome (VT1000S; Leica, Wetzlar, Germany) were prepared and sections divided into four vials. NeuN staining for neuronal nuclei was performed by Neu-N primary antibody (AB104224; Abcam, Tokyo, Japan) and a secondary antimouse IgG-conjugated biotin (Invitrogen, Tokyo, Japan). NeuN signals were enhanced by an avidin–biotin complex method (ABC Elite; Vector Laboratories, Tokyo, Japan) and visualized using a metal-enhanced DAB Substrate Kit (#34065; Thermo Scientific, Tokyo, Japan). Images of sections were obtained using a digital microscope (BZ-9000; Keyence, Osaka, Japan) and inspected for obvious qualitative signs of neuronal loss.

### 2.7. Statistical Analysis

For statistical analysis of group differences in the *ex vivo* wedge experiments, we used one-way analysis of variance (ANOVA) to test for overall group differences and Tukey’s multiple comparisons post hoc test for contrasts. For statistical analysis of group differences in the *in vivo* experiments, we used general linear model mixed model (GLMM) [30,31] analysis of data, further dissected by Fisher’s least significant difference post hoc analyses. In the *in vivo* experiments GLMM analysis was used, due to the use of multiple control groups of smaller size.

## 3. Results

### 3.1. Induction of Seizure Activity in Three Different Models

Seizure-like activity was reliably induced in all three preparations, as illustrated in Figure 2. In hippocampal brain slices, repetitive electrical stimulation of Schaffer collaterals in area CA1 in the presence of high K^+^, Mg^2+^-free conditions produced SLEs reliably in six slices from six animals. An example trace from these slices showing induction of SLEs by electrical stimulation in hippocampal area CA is shown in Figure 2A.

In wedges of entorhinal cortex, K^+^, Mg^2+^-free solution induced spontaneous SLEs. These began 40 to 120 min after switching to high K^+^, Mg^2+^-free solution. An example trace of a spontaneous SLE from this set of wedges is shown in Figure 2B. In 20 cases out of 43 wedges from 32 animals (i.e., in 47% of wedges) the SLEs spontaneously transitioned to faster, shorter, and more continuous LRDs. An example of this transition is shown in Figure 2C.

In the whole animal (Figure 2C,D), PTZ application induced seizure activity, indicated by a shift in the EEG power spectrum to higher frequencies. In particular, the power in the 0–1 Hz EEG frequency band, which is normally high under control conditions due to urethane anesthesia (Figure 2D, Baseline), decreased markedly 5 min after PTZ injection (Figure 2D, After PTZ). Conversely, the power in the 1–3 Hz EEG frequency band, which is usually very low at baseline, markedly increased after PTZ, with a prominent peak in the group average across all 18 animals.

To confirm that the muscimol release from muscimol-containing HGN-liposomes in normal brain tissue is sufficient to evoke a physiological effect, we used laser stimulation during whole-cell recording from CA1 pyramidal cells in the presence muscimol-containing HGN-liposomes in brain slices. Laser exposure caused transient hyperpolarizations of the membrane potential that were similar to GABAegic inhibitory postsynaptic potentials (Figure 3A,B). The repeatability and accuracy of the laser-induced release is shown in the repeated hyperpolarizations induced over multiple stimuli, which showed a distinct time-course with repeated stimuli (Figure 3C). The membrane potential time course indicates that the half-life of the electrophysiological effects of muscimol after release from HGN-liposomes is extremely short, on the order of seconds. Thus, our intracellular recordings of the timecourse of the response to release of muscimol from HGN-liposomes reveal a rapid, reversible, and repeatable action, on a timescale of seconds. The effects that we observed of applying small amounts locally were very fast.

### 3.2. Amelioration of Seizure Activity by Muscimol Release from Liposomes Ex Vivo

We found that muscimol release from HGN-liposome reduced seizure activity in the three seizure models. In the hippocampal brain slice, laser stimulation of HGN-liposomes that did not contain muscimol had no effect on evoked SLEs (Figure 4A). In contrast, in the presence of muscimol-containing HGN-liposomes, laser stimulation applied at the onset of the electrical stimulus train blocked the SLEs (Figure 4B). As illustrated in the example in Figure 4B, this effect was consistently evoked with repeated laser stimulation in the same slice. Analysis of pooled data across all experiments confirmed that, on average, the number of epileptiform events per minute was reduced on each occasion that laser stimulation was applied in the presence of muscimol-containing HGN-liposomes, but not with control HGN-liposome containing no muscimol (Figure 4C). Laser stimulated release of muscimol also reduced spontaneous SLE frequency (Figure 4D).

Having established the efficacy of laser induced muscimol release from HGN-liposome for reducing electrical-stimulation induced seizure activity, we then measured the effect of ultrasound release of muscimol on the frequency of spontaneous SLEs. Figure 5A shows a representative recording from an entorhinal-wedge expressing spontaneous epileptiform activity under Mg^2+^-free conditions. Exposure of the preparation to ultrasound-stimulated HGN-liposome was associated with a dramatic reduction in SLE frequency. In contrast, wash-in of unstimulated muscimol-containing HGN-liposomes caused little change. Group analysis (Figure 5B) revealed an overall effect of treatment (one-way ANOVA, *p* < 0.0001, F (2,25) = 21.0). Post hoc tests confirmed a significant reduction in the frequency of epileptiform activity after the application of ultrasound to the formulation, compared to both the pre-ultrasound wash in period (*p* < 0.0001; Tukey’s multiple comparisons post hoc test) and to the equivalent post-wash in period with no US applied (*p* = 0.0035). There was no significant difference between the wash-in and post-wash-in, no US periods.

### 3.3. Amelioration of Seizure Activity by Muscimol Release from Liposomes In Vivo

The foregoing experiments indicated that seizure activity could be arrested by release of muscimol from HGN-liposomes *ex vivo*. In order to determine whether the treatment would be effective *in vivo*, experiments were conducted in anesthetized whole animals expressing seizure activity induced by PTZ applications. HGN-liposomes were intravenously injected and ultrasound stimulation was applied via an extradural collimator (Figure 1C). Four groups were compared (PTZ, PTZ plus HGN-liposome, US, and PTZ plus HGN-liposome plus US). The key finding of these experiments concerned the effect of US-activated release of muscimol on the PTZ-induced seizure signatures in the EEG.

As shown in Figure 6, before any treatment, all groups showed changes in the power of the EEG post-PTZ (5–9 min period) relative to the pre-PTZ baseline (0 min) in all EEG frequency bands. In the 0 to 1 Hz EEG band (Figure 6A) there was a decrease, with increases in the 1 to 3 Hz and 3–5 Hz EEG bands (Figure 6B,C). These effects of PTZ treatment, reflecting successful generation of the model, were confirmed by statistical analyses using GLMM procedures, which revealed significant main effects of time in all bands (0 to 1 Hz, F (1,31) = 63.49, *p* < 0.001; 1 to 3 Hz, F (1,31) = 177.24, *p* < 0.001; 3–5 Hz, F (1,31) = 29.07, *p* < 0.001, respectively).

As expected, before treatment there was no significant difference in the effect of PTZ between rats allocated to receive either US application to muscimol-containing HGN-liposomes (blue lines and symbols in Figure 6A,B; *n* = 8); muscimol-containing HGN-liposome without US application, (black lines and symbols; *n* = 3); ultrasound application only, with no liposomes present (green lines and symbols; *n* = 4); or PTZ-only, with no subsequent manipulation (red lines and symbols; *n* = 3); overall ANOVA by EEG frequency bands (0–1Hz, F (3,31) = 1.97, *p* = 0.39, 1–3 Hz F (3,31) = 1.94, *p* = 0.144, 3–5 Hz F (3,31) = 0.401, *p* = 0.753), GLMM. This shows that the experimental and control groups were not significantly different in the way they responded to PTX prior to treatment.

After treatment, effects differed by group and EEG bands. For the 0–1 Hz EEG band, in the group that received US application to muscimol-containing HGN-liposomes (blue lines and symbols in Figure 6A,B; *n* = 8), the EEG power reverted towards pre-PTZ levels. In contrast, there was little change over the same time period in the control groups. These control groups received either, application of muscimol-containing HGN-liposome without US application, (black lines and symbols; *n* = 3); ultrasound application only, with no liposomes present (green lines and symbols; *n* = 4); or PTZ-only, with no subsequent manipulation (red lines and symbols; *n* = 3). Similar effects were seen in the 1–3 Hz band (Figure 6B). These differential effects led to a visible separation of the traces for the test group versus the controls. There was no visible effect in the 3–5 Hz band (Figure 6C). These effects were confirmed by the GLMM analyses of post-PTZ data, which for both 0–1 and 1–3 Hz EEG bands revealed a significant main effect of group (0–1 Hz, F (3) = 5.007, *p* = 0.004; 1–3 Hz, F (3) = 2.907, *p* = 0.043), but not for the 3–5 Hz band (F (3) = 2.387, *p* = 0.079). There were no main effects of time, and no significant group x time interactions. The significant effects were further dissected using Fisher’s least significant difference post hoc analyses. For the 0–1 Hz band, Group 4 (treatment with muscimol-containing HGN-liposomes + ultrasound) was significantly different from all other groups (*p* < 0.022 in all cases), while there were no differences between the other groups. For the 1–3 Hz band, Group 4 was significantly different from group 2 (US only; *p* = 0.02) and group 3 (muscimol-containing HGN-liposomes only; *p* = 0.026), but the contrast with group 1 (no treatment) did not reach significance (*p* = 0.126).

### 3.4. Unstimulated HGN-Liposome Effectively Sequester Bioactive Compounds In Vivo

One of the potential advantages of HGN-liposome drug delivery is that drugs can be sequestered in the body without activity, until released on demand. That this occurs is suggested by the lack of effect of infusion of muscimol-containing HGN-liposome in the absence of ultrasound stimulation (Figure 5 and Figure 6). To further test the ability of the HGN-liposome used here to sequester bioactive compounds, we loaded HGN-liposome with a neurotoxin, kainic acid (KA), which causes widespread cell death when injected into brain, and injected it into the cerebral cortex. Following one week of survival we examined the effects of the injection on neural tissue. As shown in Figure 7, injection of HGN-liposome containing KA had no effect (Figure 7B), whereas injection of unencapsulated KA solution caused damage to neural tissue, indicated by the pale areas in Figure 7C. Together, these results indicate that the HGN-liposome used here are able to sequester drugs until release is triggered.

## 4. Discussion

The main finding of this study was that on-demand release of muscimol using a HGN-liposome drug delivery system effectively reduced seizure activity in *ex vivo* and *in vivo* experimental models. In brain slices, patch-clamp recordings from hippocampal pyramidal cells showed that laser-triggered release of the GABA agonist muscimol from HGN-liposome caused hyperpolarization of the membrane potential and blockade of action potentials. This remotely-controlled release of muscimol also arrested seizure activity (SLEs, LRDs) in the hippocampus and entorhinal cortex wedges. In anaesthetized animals, *in vivo* US stimulation of previously intravenously injected HGN-liposomes caused attenuation of PTZ-induced SLEs, observed as increased power in the high frequency spectrum of the EEG. Finally, HGN-liposomes protected brain tissue from damage by intra-liposome neurotoxin for at least one week, demonstrating effective containment. Together, these findings demonstrate the potential for effective reduction of seizure activity *in vivo*, with reduced toxicity, by remotely controlled release of drugs sequestered in HGN-liposome.

We used muscimol in these proof-of-concept experiments, rather than approved medications, for several reasons. First, we were aiming for immediate seizure suppression, on demand, and focally at the site of seizure generation. Muscimol is both potent and rapidly-acting, thus is suitable for this approach. As shown in the present paper, and earlier studies [12,13,14,15], muscimol is immediately effective in attenuating seizures when applied locally at the site of the seizures. In contrast, the approved drugs are optimized for systemic treatment with minimal side effects, require several dosing cycles to make them effective at stopping seizures, and do not act as quickly when locally applied. Thus, because they are not optimal for local application and generally effective given systemically, there is less value in using them in HGN-liposome delivery. Second, about one-third of people with epilepsy have seizures refractory to systemic pharmacotherapy with approved medications [32]. For these people in particular, new approaches are needed. Third, as noted by Gernert [32], targeted intracranial delivery, by providing higher drug concentrations in localized target regions, and lower concentrations in other brain or peripheral areas, allows the use of drugs that are otherwise unsuitable for systemic administration because of their toxicity or poor uptake into the brain. Since intracranial delivery of muscimol, in small quantities, has been shown to be safe in previous studies [11,12,13,14,15,18,19,20], we used it as a test of the delivery system.

In the present study, we used laser stimulation to cause release of muscimol from HGN-liposomes in brain slices. Several previous studies have established that substances can be released from HGN-liposome nanostructures on a rapid timescale by laser stimulation in non-biological assays [27,33,34,35]. These studies demonstrated the feasibility of drug delivery on a rapid timescale using laser stimulation. We have also previously shown that release and dosage can be controlled by varying the number and intensity of femtosecond pulses of light [27]; and, furthermore, that on-demand release of different neurochemicals and drugs from HGN-liposome in live brain tissue has rapid, repeatable, and reliable physiological effects [28]. However, at present, laser-stimulated release from HGN-liposomes is not suitable for *in vivo* use, because light does not penetrate far through the skull or brain parenchyma, and miniature femtosecond pulsed lasers are not available for chronic implantation. On the other hand, focused US can be transmitted through the skull and brain tissue. Recent work has demonstrated the feasibility of US-stimulated release of drugs from liposomes *in vivo* [2]. We have also shown that in vitro, US can evoke multiple release events of a constant amount over 25 individual applications [6]. The present study extends this work, by showing that transcranial US-stimulation, both *ex vivo* and *in vivo*, can cause sufficient release of muscimol from HGN-liposomes, to arrest ongoing seizure activity.

Our intracellular recordings of the timecourse of the membrane potential hyperpolarization show that the timecourse of the inhibitory effect of muscimol after release from HGN-liposomes is extremely short, on the order of seconds. This finding is consistent with previous reports of rapid, subsecond to second clearance of other neurotransmitters such as dopamine [27] and glutamate [28], measured after release from laser-stimulated HGN-liposomes. The amount of release and, hence, peak concentration of the drug obtained with each stimulation is linearly related to duration and intensity for both laser [27] and US [6] stimulation. Since only a small percentage of content is released with each stimulus, dose can be titrated against clinical effect by increasing intensity or duration of stimulation, and terminated immediately by turning off stimulation. These properties of the drug delivery system provide a means for precise control over drug actions.

The very small quantities of muscimol contained in, and released from, HGN-liposomes are unlikely to cause adverse effects. The amount of muscimol needed to produce therapeutic effects by local application in the brain is small compared to the amount that would cause side effects after diffusion into the cerebrospinal fluid and distribution throughout the bloodstream. Studies in non-human primates showed that administration of muscimol into the subarachnoid space suppressed seizures locally, but otherwise led to no detectable levels of muscimol in blood or cisternal CSF [15]. Delivery of 1.0–2.5 mM muscimol into the neocortex of rodent and nonhuman primate models has been shown to have powerful antiepileptic effects, without adverse effects on the animal’s behavior [13,36]. Muscimol is metabolized in the brain and periphery and excreted in the urine in roughly equal proportions as unchanged muscimol, a cationic conjugate, or an oxidation product [9,37]. Experiments with [3H] muscimol showed that it rapidly disappears from the blood [8].

The biocompatibility, distribution and eventual fate of the liposome constituents and HGNs is less clear and possibly more of a concern than the distribution of muscimol. After intravenous injection liposomes in circulation might be sequestered in liver or spleen. They might also cause immune reactions peripherally or cellular changes after crossing the BBB. Some of the pitfalls have been reviewed recently [38]. Gold, the constituent of the HGN component, has been used medically in ionic form in the treatment of human rheumatoid arthritis, and the literature concerning adverse reactions to ionic gold has been reviewed recently [39]. In rats, gold nanoparticles were found to be biocompatible and relatively innocuous after intravenous injection, but the highest accumulation was in spleen and lowest in brain [40]. Laser-synthesized gold nanoparticles are considered to be purer and safe for biomedical applications [41], without causing liver damage. However, studies of gold nanoparticle effects in mice have revealed an increased rate of abortion and fetal abnormalities if given in the early pregnancy [42]. Reviews of this topic highlight the limited available evidence and need for more knowledge concerning the toxicity of HGN after injection [43]. Excretion of accumulated particles from the liver and spleen can take up to 3 to 4 months, indicating that further studies of the toxicity of HGNs are needed.

Even if biocompatibility issues can be overcome, clinical application of muscimol-containing HGN-liposomes will not be feasible until future technological developments provide a practical means to infiltrate them into the brain parenchyma. In the present study, HGN-liposomes may have gained access to the brain as a result of the seizure activity itself [44], or by momentary disruption of the BBB by the US stimulation, allowing liposome penetration [45]. The combination of systemic injection of HGN-liposomes with focal US stimulation might, thus, achieve a high local concentration of muscimol in the brain or brain vasculature, with relatively low concentration in the periphery, due to encapsulation within liposomes and dilution of the cerebrally released muscimol. However, procedures such as carotid or intracerebral injections, as used in the present study, are invasive neurosurgical procedures that might only be considered in intractable drug resistant epilepsy [32]. For routine use, less invasive methods will be required to move HGN-liposomes across the BBB and into the brain. Trojan horse liposomes (THLs) may be a future possibility. THLs are pegylated liposomes with a receptor-specific monoclonal antibody targeted to receptors that can transport liposomes across the blood–brain barrier, such as the transferrin receptor [46,47]. The antibody is conjugated to the surface of the THL and the transferrin receptor ferries the liposome across the BBB. Further work is needed to determine if HGN-liposomes can be transported intact from the blood into the brain by hijacking existing transport mechanisms.

To be considered as a possible treatment, liposomes injected *in vivo* must also retain the ability to release their contents over a useful time period following administration. Previous work measuring release in brain slices from previously injected animals has shown that liposomes retain their ability to repeatedly release drug after one week *in vivo* [28]. Further work is needed to determine the time course over which liposomal nanostructures remain intact and responsive to ultrasound stimulation after injection *in vivo*. Another challenge is to minimize leakage from liposomes, while preserving the ability to trigger release. In the current work, we show that the HGN-liposomes successfully encapsulated KA and could be injected into the brain and remain there for one week, without causing damage to surrounding tissue. Leakage can be minimized by increasing the stability of the liposome formulation. However, there is a trade-off between the stability of liposomes and sensitivity to stimulation, which requires systematic study to arrive at optimal formulations.

Ideally, treatment for epilepsy should prevent seizures before they occur. However, in one third of patients existing treatments are not effective in preventing seizures, and treatment-resistant epilepsy is associated with significant morbidity and mortality [48,49]. Existing technology is not yet capable of predicting seizures with clinically useful reliability. However, the technology for seizure detection already exists. For example, Kim et al. [50] concluded from a review of the literature that ‘… the state-of-the-art seizure detection system performance is sufficient to build a robust and reliable wearable device that could be used for daily seizure monitoring and classification.’ What is needed, therefore, is a means to deliver the drug immediately on the first sign of a seizure. Here we aimed to demonstrate proof of principle that seizures can be arrested almost instantaneously provided HGN-liposomes are preloaded in the brain. However, the practical use of this approach will require the future development of tools for applying US or laser stimulation in an ambulatory patient setting, as well as new technology for infiltrating the HGN-liposomes into the brain. Extensive studies of pharmacokinetics, pharmacodynamics, and toxicity will also be needed to determine the utility and safety of the technology in humans.

## 5. Conclusions

The present experiments demonstrated that the HGN-liposome formulation we have developed is able to encapsulate and contain muscimol, and release it in the brain in response to femtosecond laser or US stimulation. Release is rapid and immediate, causing fast and repeatable hyperpolarizations of neurons, similar to physiological inhibitory postsynaptic potentials. In *ex vivo* seizure models, stimulation of muscimol loaded HGN-liposomes caused immediate suppression of spontaneous and electrically-evoked seizure activity. Experiments also showed that ultrasound stimulation applied to the brain through the dura attenuated seizure activity induced by PTZ in rats given intravenous injection of muscimol-loaded HGN-liposomes. We also showed that intracerebrally injected HGN-liposomes loaded with toxic concentrations of KA did not cause damage to surrounding tissue. Thus, we demonstrated the feasibility of precise temporal control over exposure of neurons to the drug, potentially enabling therapeutic effects without continuous exposure. Overall, these findings suggest that HGN-liposomes combined with ultrasound triggering have potential for the development of innovative treatment strategies for neurological disorders, using on-demand release of pharmaceuticals.

The present study focused on epileptic seizures in particular, because of the challenges of long-term treatment with systemic antiepileptic drugs, and the large number of patients with drug-resistant epilepsy. The ability to deliver high concentrations of drug to target areas on demand, while keeping drug concentrations low at other sites and times may enable the use of drugs that are effective when applied locally, but unsuitable for systemic use, because of their effects on other systems. Muscimol is one example of such a drug, which has been found unsuitable for systemic application, but potentially effective when applied locally. For such applications, the development of technology to move HGN-liposomes across the BBB and anchor them with the brain parenchyma would be necessary. Further work is needed to determine the utility and safety of the technology in humans, particularly concerning the pharmacokinetics, pharmacodynamics, and toxicity of HGN-liposomes and their constituents. Technology for detection of seizures and application of US stimulation in ambulatory patients will also be needed. If these problems can be solved, HGN-liposomes have the potential to be developed into a new treatment for responsive forms of epilepsy.

## Figures and Tables

**Figure 1 pharmaceutics-14-00468-f001:**
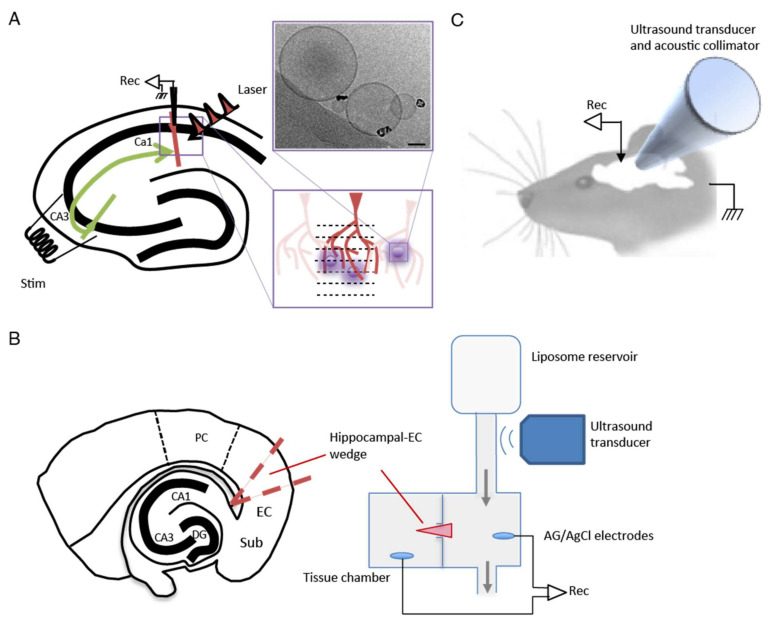
Experimental setups for three different models. (**A**) Hippocampal slice experiments. Schematic of hippocampal slice shows location of electrical field stimulating electrode (Stim) in area CA3, and whole cell intracellular recording electrode (Rec) in area CA1. Femtosecond pulses (Laser) were applied via the use of a 2-Photon microscope. Expanded schematic of CA1 shows location of dendrites (red) and liposomes (purple) in relation to sequential laser scan lines (dashes), forming a grid pattern in the dendritic zone. Example cryo-transmission electron microscope image shows liposomes tethered to hollow gold nanoshells (black; scale bar = 50 nm). (**B**) Entorhinal cortex wedge experiments. Schematic shows position of entorhinal cortex wedge, and wedge positioned in two-compartment grease gap chamber. Liposomes were stimulated with ultrasound as they passed into the tissue chamber. Grey arrows indicate flow of ACSF. (**C**). *In vivo* preparation. Field potential signals were obtained via an extracellular recording electrode (Rec) in the left frontal cortex. Liposomes were stimulated using an ultrasound transducer coupled to an acoustic collimator and positioned on the dura above the cortex.

**Figure 2 pharmaceutics-14-00468-f002:**
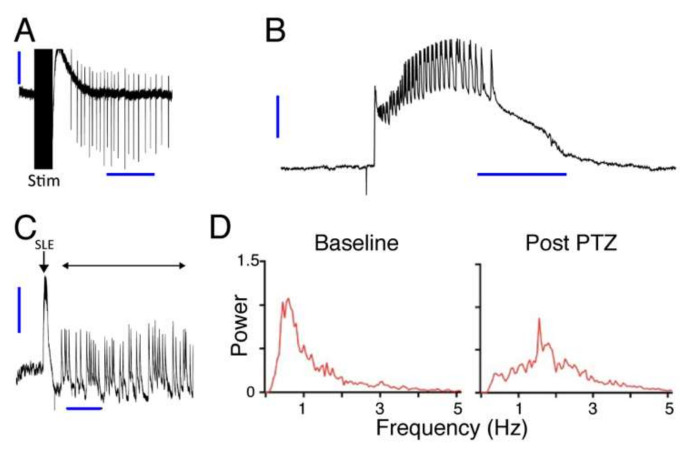
Example data from experiments showing induction of SLEs in three different models. (**A**) *Ex vivo* hippocampal slice preparation. Trace shows field potentials recorded from area CA1. Repetitive electrical stimulation (Stim) induced clonic afterdischarges. Vertical and horizontal scale bars (thick blue lines) show 0.2 mV and 5 s. (**B**,**C**). *Ex vivo* entorhinal cortex grease gap recordings. (**B**) Trace shows spontaneous SLE after removal of extracellular Mg^2+^ from the perfusing ACSF. Vertical and horizontal scale bars (thick blue lines) show 0.5 mV and 20 s. (**C**) Longer timescale trace shows transition of SLE (arrow) to late recurrent discharges (double headed arrow). Vertical and horizontal scale bars (thick blue lines) show 0.2 mV and 5 min. (**D**) *In vivo* anaesthetized animal preparation. Traces show EEG power spectrum before (Baseline) and after (Post PTZ) application of PTZ. Frequency (Hz) in D refers to EEG frequency.

**Figure 3 pharmaceutics-14-00468-f003:**
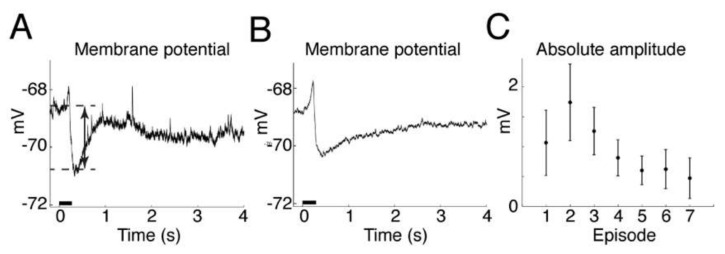
Effects of laser-induced muscimol release on hippocampal CA1 pyramidal neuron membrane potential measured in whole-cell recording, current-clamp mode. (**A**) Trace shows single trial example. Laser stimulation (horizontal black bar) induces transient hyperpolarization. Double-headed arrow indicates amplitude of hyperpolarization measured between baseline and peak average values. (**B**) Average of first 10 episodes from same neuron, as in (**A**). (**C**) Graph shows absolute value of group average hyperpolarization responses to first seven laser stimulation episodes (*n* = 3 animals, mean ± SEM).

**Figure 4 pharmaceutics-14-00468-f004:**
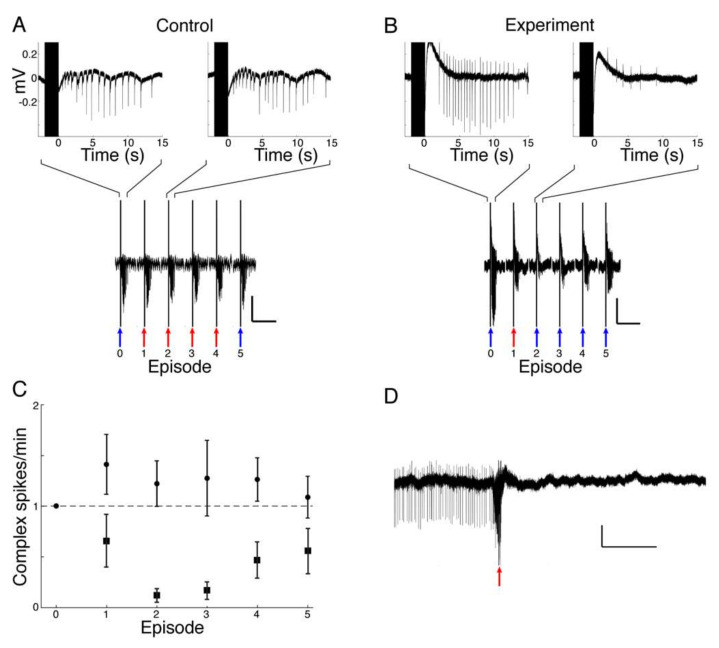
Effect of laser-stimulated release of muscimol on SLEs recorded in hippocampal area CA1. (**A**) No muscimol control. Top traces, two typical responses to electrical stimulation. Lower trace shows the series of six successive SLE-inducing electrical stimuli on a compressed time scale, from which the top traces are taken (Scale bars show 0.2 mV and 1 min). Blue arrows show episodes where only electrical stimulation was applied. Red arrows show episodes where electrical and laser stimulation was combined. In the presence of liposomes containing no muscimol, repeated laser stimulation had no effect on electrical stimulation-evoked SLE. (**B**) Effect of laser-stimulated muscimol release on SLE. Conventions, as for A, in the presence of muscimol-containing HGN-liposomes. Note the reduction in electrically-evoked complex spikes after muscimol release. (Scale bars show 0.2 mV and 1 min). (**C**) Group average baseline normalized complex spike activity in control (circles, *n* = 3 animals) and experimental (squares, *n* = 3 animals) conditions. (**D**). Effect of laser-induced muscimol release (red arrow) on spontaneous SLE. Trace shows example in which spontaneously occurring complex spikes (thin lines) were eliminated by laser release of muscimol (red arrow). Scale bars in (**A**,**B**,**D**) show 0.1 mV and 10 s.

**Figure 5 pharmaceutics-14-00468-f005:**
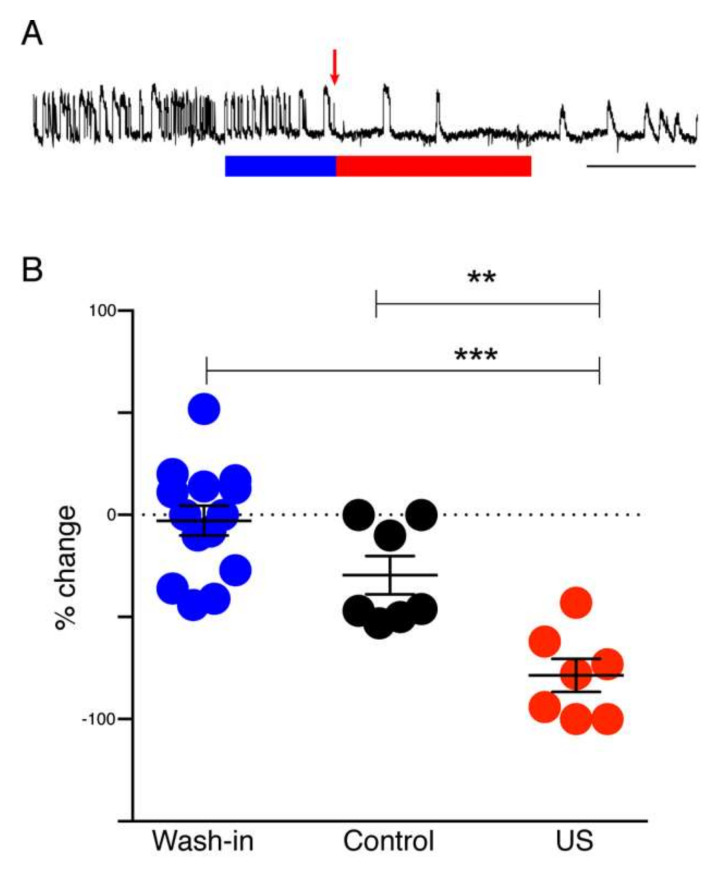
Effect of ultrasound-stimulated release of muscimol on SLEs in hippocampal–entorhinal wedge preparation. (**A**) Example recording shows typical reduction of seizure activity following ultrasound stimulation. Blue bar, wash-in period of unstimulated muscimol-containing liposomes. Red arrow, time of ultrasound stimulation of liposomes in the reservoir supplying ACSF to the tissue chamber. Red bar, duration of subsequent exposure to solution containing ultrasound stimulated liposomes. Scale bar, 15 min. (**B**) Graph shows group mean ± standard error for percent change of seizure activity (SLE + LRD). Filled circles show individual animal data. Blue circles, wash-in period data; black circles, post wash-in data from no-ultrasound stimulation controls; red circles, post wash-in data from ultrasound stimulated slices (US). ** *p* < 0.01 and *** *p* < 0.0001, Tukey’s post hoc comparisons.

**Figure 6 pharmaceutics-14-00468-f006:**
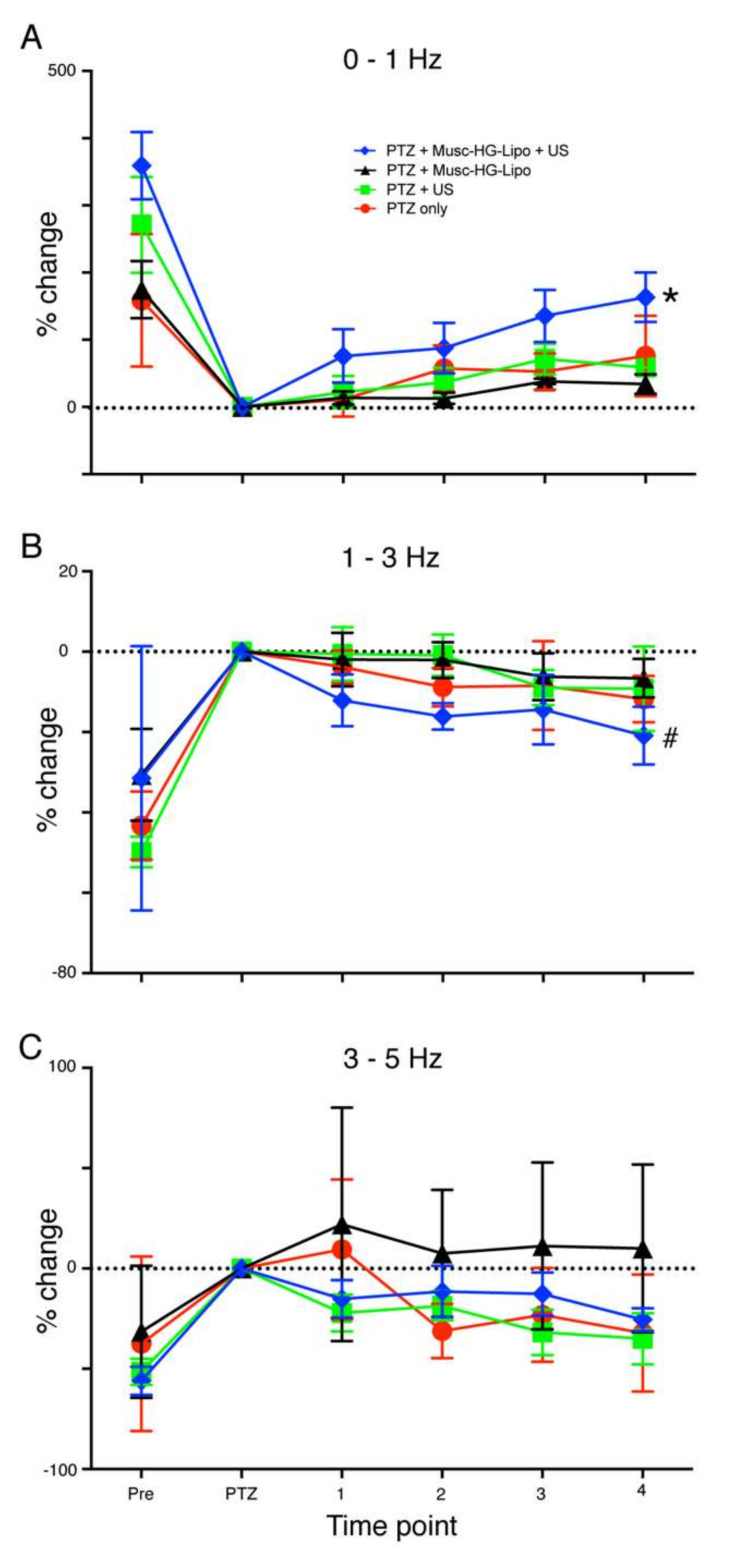
Ultrasound stimulation of liposomes reduces seizure activity *in vivo*. Line plots show effect of experimental and control treatments on EEG power measured in different EEG frequency bands. In all groups PTZ treatment causes a significant difference from pre-treatment measures. (**A**) * indicates significant main effect of treatment (*p* = 0.004) on power measured in the 0–1 Hz EEG band. Post hoc contrasts showed that the group that received US application to muscimol-containing HGN-liposomes (PTZ + Musc-HG-Lipo + US, blue line and symbols, *n* = 8) was significantly different from all control groups (*p* < 0.022 in all cases), while there were no differences between the control groups that received either, application of muscimol-containing HGN-liposome without US application (PTZ + Musc-HG-Lipo, black line and symbols; *n* = 3); ultrasound application only with no liposomes present (PTZ + US, green lines and symbols; *n* = 4); or PTZ-only with no subsequent manipulation (PTZ only, red lines and symbols; *n* = 3). (**B**) # indicates significant main effect of treatment in 1–3 Hz EEG band (*p* = 0.043). In this band the group that received US application to muscimol-containing HGN-liposomes was significantly different from the group that received US only (*p* = 0.02) and the group treated with muscimol-containing HGN-liposomes only (*p* = 0.026), but the contrast with the group receiving no treatment did not reach significance (*p* = 0.126). (**C**) There was no significant effect of treatment on power measured in the 3–5 Hz band (*p* = 0.079). Data are normalized to power measured over 5 min, following PTZ administration and expressed as percentage change. See text for details of statistical analysis.

**Figure 7 pharmaceutics-14-00468-f007:**
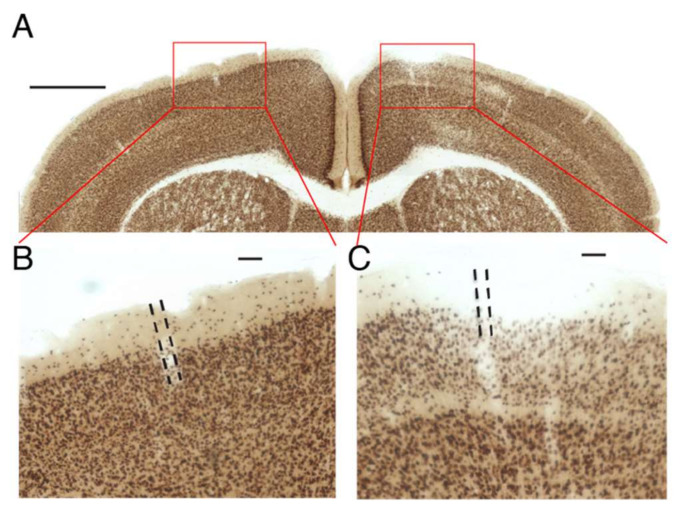
Liposomes effectively sequester drugs *in vivo*. (**A**) Transverse section of brain stained for cell bodies (NeuN), showing injection sites of KA-containing liposomes (**left**) and unencapsulated KA solution (**right**). Red rectangles show location of enlarged images in (**B**,**C**). Scale bar, 1 mm. (**B**,**C**). Enlargements of sites shown in (**A**). Dashed lines indicate injection tracks. Scale bar 100 µm.

## Data Availability

Data is contained within the article and Appendix A.

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
