# Peer review of "An On-Demand Drug Delivery System for Control of Epileptiform Seizures"

_pharmaceutics, 2022, doi:10.3390/pharmaceutics14020468_

Round 1
Reviewer 1 Report
Dear Authors,
The publication is very interesting. However, I noted the lack of studies (in vitro) on the release of the drug substance. If it is possible, please complete them. Moreover, abstract does not reflect the content of the article.
Best regards,
Reviewer
Author Response
Response to Reviewers
We thank the reviewers for their careful reading of the manuscript and helpful comments. We have undertaken major revisions to address all the points of the referees, and we believe the paper is substantially improved as a result of their input.
Reviewer 1
Reviewer Comment
The publication is very interesting. However, I noted the lack of studies (in vitro) on the release of the drug substance. If it is possible, please complete them.
Author response
We have previously reported release measurements of dopamine, as well as carboxyfluorescein, from ultrasound and laser activated HGN liposomes [1, 2]. In the present study we were unable to do in vitro release studies for muscimol because we couldn’t find a chemical, electrochemical, or spectroscopic assay that could distinguish between encapsulated and non-encapsulated muscimol and that was sensitive enough to measure its release in real time. Hence, we used the in vivo biological assay as described. This is now described in the Introduction (lines 153-160).
Reviewer Comment
Moreover, abstract does not reflect the content of the article.
Author response
We have rewritten the abstract to reflect the content of the article, as follows:
“Abstract
Drug delivery systems have the potential to deliver high concentrations of drug to target areas on demand, while elsewhere and at other times encapsulating the drug to limit unwanted actions. Here we show proof of concept in vivo and ex vivo tests of a novel drug delivery system based on hollow-gold nanoparticles tethered to liposomes (HGN-liposomes), which become transiently permeable when activated by optical or acoustic stimulation. We show that laser or ultrasound simulation of HGN-liposomes loaded with the GABAA receptor agonist, muscimol, triggers rapid and repeatable release in sufficient concentration to inhibit neurons and suppress seizure activity. In particular, laser-stimulated release of muscimol from previously injected HGN-liposomes caused subsecond hyperpolarizations of the membrane potential of hippocampal pyramidal neurons, measured by whole cell intracellular recordings with patch electrodes. In hippocampal slices and hippocampal-entorhinal cortical wedges, seizure activity was immediately suppressed by muscimol release from HGN-liposomes triggered by laser or ultrasound pulses. After intravenous injection of HGN-liposomes in whole anesthetized rats, ultrasound stimulation applied to the brain through the dura attenuated seizure activity induced by pentylenetetrazol. Ultrasound alone, or HGN-liposomes without ultrasound stimulation, had no effects. Intracerebrally injected HGN-liposomes containing kainic acid retained their contents for at least one week without damage to surrounding tissue. Thus, we demonstrate the feasibility of precise temporal control over exposure of neurons to the drug, potentially enabling therapeutic effects without continuous exposure. For future application, studies on the pharmacokinetics, pharmacodynamics, and toxicity of HGN-liposomes and their constituents, together with improved methods of targeting, are needed to determine the utility and safety of the technology in humans.”
Reviewer 2 Report
This work describes an on-demand anti-epilepsy treatment based on liposomes.
While the authors convinced me about the quality of their delivery system and the ability to trigger the drug release via external stimuli (laser or ultrasound), this paper sounds too much of translational while the path to clinics is still very long.
In particular, I criticize:
1) the concept of on-demand: In my opinion, this means that during a seizure the liposomes are triggered by some changes in the neuron potential and respond by releasing the drug. This cannot be the case of laser or US triggering. One could think of a tool applied on the skull that detecting the seizure send US or laser stimuli. But this is more science fiction right now.
2) The second thing I am not satisfied is the experimental setting. To be on-demand the liposome need to quietly staying in the brain and release their payload in response to the stimulus.
On the other hand, the authors induce the seizure and inject the liposomes in the carotids applying the external stimulus. This is a kind of unlikely in real life.
In conclusion, I think that at the moment the paper should be rejected, but if the author tune considerably down the future application of the system and will focus more on the data, I believe this is very decent work and this platform might have in the future some translational potential.
Author Response
Response to Reviewers
We thank the reviewers for their careful reading of the manuscript and helpful comments. We have undertaken major revisions to address all the points of the referees, and we believe the paper is substantially improved as a result of their input.
Reviewer 2
Reviewer Comment
This work describes an on-demand anti-epilepsy treatment based on liposomes.
While the authors convinced me about the quality of their delivery system and the ability to trigger the drug release via external stimuli (laser or ultrasound), this paper sounds too much of translational while the path to clinics is still very long.
Author response
We agree with the reviewer and have extensively revised the paper, emphasizing the data and making it clearer that this is a proof of concept, electrophysiological study, and the path to potential clinical applications is long. Due to the extensive nature of the changes it is not practical to list them here. We have also completely rewritten the abstract to address this issue (new version copied below):
Abstract
Drug delivery systems have the potential to deliver high concentrations of drug to target areas on demand, while elsewhere and at other times encapsulating the drug to limit unwanted actions. Here we show proof of concept in vivo and ex vivo tests of a novel drug delivery system based on hollow-gold nanoparticles tethered to liposomes (HGN-liposomes), which become transiently permeable when activated by optical or acoustic stimulation. We show that laser or ultrasound simulation of HGN-liposomes loaded with the GABAA receptor agonist, muscimol, triggers rapid and repeatable release in sufficient concentration to inhibit neurons and suppress seizure activity. In particular, laser-stimulated release of muscimol from previously injected HGN-liposomes caused subsecond hyperpolarizations of the membrane potential of hippocampal pyramidal neurons, measured by whole cell intracellular recordings with patch electrodes. In hippocampal slices and hippocampal-entorhinal cortical wedges, seizure activity was immediately suppressed by muscimol release from HGN-liposomes triggered by laser or ultrasound pulses. After intravenous injection of HGN-liposomes in whole anesthetized rats, ultrasound stimulation applied to the brain through the dura attenuated seizure activity induced by pentylenetetrazol. Ultrasound alone, or HGN-liposomes without ultrasound stimulation, had no effects. Intracerebrally injected HGN-liposomes containing kainic acid retained their contents for at least one week without damage to surrounding tissue. Thus, we demonstrate the feasibility of precise temporal control over exposure of neurons to the drug, potentially enabling therapeutic effects without continuous exposure. For future application, studies on the pharmacokinetics, pharmacodynamics, and toxicity of HGN-liposomes and their constituents, together with improved methods of targeting, are needed to determine the utility and safety of the technology in humans.
Reviewer Comment
1) the concept of on-demand: In my opinion, this means that during a seizure the liposomes are triggered by some changes in the neuron potential and respond by releasing the drug. This cannot be the case of laser or US triggering. One could think of a tool applied on the skull that detecting the seizure send US or laser stimuli. But this is more science fiction right now.
Author Response
We agree that tools for applying US or laser stimuli in an ambulatory patient setting are not yet developed, and we acknowledge this limitation in the Discussion (Lines 607–612), as copied below:
“However, the practical use of this approach will require the future development of tools for applying US or laser stimulation in an ambulatory patient setting, as well as new technology for infiltrating the HGN-liposomes into the brain.”
Reviewer Comment
2) The second thing I am not satisfied is the experimental setting. To be on-demand the liposome need to quietly staying in the brain and release their payload in response to the stimulus. On the other hand, the authors induce the seizure and inject the liposomes in the carotids applying the external stimulus. This is a kind of unlikely in real life.
Author Response
We agree that the forward aim should be to have the liposomes sequestered in the brain so that they can be activated on demand. For proof of principle we first needed to show that the liposomes, once in the brain, could be made to release muscimol. We agree that injecting the liposomes immediately prior to the ultrasound activation is unlikely to happen in real life. However, we have made an incremental step by showing that if the liposomes are in the brain by some means then they can quietly stay in the brain until stimulated, and then release their cargo. This finding motivates developing a method to infiltrate the liposomes into the brain. We have added two paragraphs of text addressing these points to the Discussion (lines 565-583 and 584-596).
Reviewer Comment
In conclusion, I think that at the moment the paper should be rejected, but if the author tune considerably down the future application of the system and will focus more on the data, I believe this is very decent work and this platform might have in the future some translational potential.
Author Response
We have extensively revised the text to tone down claims about future applications and now focus more on the data and the limitations of the experiments to date. We thank the reviewer for these suggestions, which have improved the paper.
Reviewer 3 Report
The authors conducted sophisticated experiments using advanced solutions.
However, many issues raise doubts.
minor issues:
1. The authors did not use line numbering, which makes it difficult to explain by the reviewer where the errors are.
2. At the beginning, the authors explain the abbreviations (PTZ, HGN, ACS). Why only these? There are many others (SLE, LRD, GLM, KA, ACSF, etc.). Maybe it is not necessary in this place, only in text.
3. Figure descriptions should be on the same page as figures.
4. Figure 2 A, B, C - no scale, units, no explanations. The drawings are small, while in figure 3 the graphics are too large. You can make them smaller and put them side by side — the same style for all figures.
5. "In hippocampal brain slices repetitive electrical stimulation of Schaffer collaterals in area CA1 in the presence of high K +, Mg2 + -free conditions produced SLEs (6 slices from 6 animals, Figure. 2A)." "In wedges of entorhinal cortex, Mg2 + -free solution induced spontaneous SLEs (43 slices from 32 animals, Figure. 2B)." What do the graphs in Figures 2A, 2b show if "6 slices from 6 animals", "43 slices from 32 animals" are described?
6. Figure 3A no description of the scale
7. figure 6. It is not explained what "time point" means. No statistical significance was noted. Were the groups differ from each other? The term "Muscisomes" is not explained in the text or in the figure.
8. If at the beginning of the "Results" section the authors refer to Figure 6, it should be shown here, not 5 pages later.
9. "Amelioration of seizure activity by muscimol release from liposomes in vitro" If this is a chapter name, it should be italic + period.
major issues:
1. The critical question that arises is why the authors decided to use muscimol? We only learn from the work that it is "(GABA) agonist, muscimol". Information about this substance must be added. Why this substance was chosen, mechanism of action, duration of action, elimination. How was the dose selected? Has the accuracy and repetitiveness of the muscimol release? How many times it can be released?
2. The purpose of the study is not to test the effectiveness of the new compound but the new method of its administration. So it seems obvious that the best choice at this point would be proven drugs used in the treatment of seizures (diazepam, clobazam) or progabide - approved for either monotherapy or adjunctive use in the treatment of epilepsy. These medications are safe and effective. Muscimol is a potent, selective agonist for the GABAA receptors but displays sedative-hypnotic, depressant and hallucinogenic psychoactivity as well. Next, it can produce euphoria, dream-like (lucid) state of mind, out-of-body experiences and synesthesia, negative effects include mild to moderate nausea, stomach discomfort, increased salivation and muscle twitching or tremors. In large doses strong dissociation or delirium may be felt. Were risk of those side effects were examined?
Muscimol was in clinical trial phase I for epilepsy, but the trial was discontinued.
Lonser, Russell R.; Oldfield, Edward H.; Sato, Susumu; Rene' Smith, R. N.; Walbridge, Stuart; Heiss, John D. (2012-08-01). "174 Convection-Enhanced Delivery of Muscimol to the Epileptic FocusPreclinical and Clinical Research". Neurosurgery. 71 (2): E568. doi:10.1227/01.neu.0000417764.02569.dc. ISSN 0148-396X.
3. Effects of muscimol last for 5–10 hours because it is not metabolized but eliminated in urine. If you prepare "on-demand drug-delivery system" and it should be activated before (?) sizers, it should end its action rapidly as well. If not, how can we control its action? What if overdose?
A good example of a treatable drug is using remifentanil in anesthesia. Its action takes 2-5 minutes so this feature is very useful in regulating deepness of action.
4. The authors described epilepsy as an ideal candidate for "drug-on-demand" treatment. An epilepsy attack is usually unexpected and unpredictable. How would the patient release the drug before a seizure? If seizures do occur, the patient is usually unconscious or unable to act. If the seizure occurrence were to be a triggering factor, then the drug would not prevent seizures from occurring but would only act symptomatically when seizures occur. The goal of the currently approved as appropriate treatment of epilepsy is to take medications to PREVENT the occurrence of seizures, as their presence increases the risk of microdamage in the brain and the severity of epilepsy in the future. Treatment that does not provide protection but only treats existing seizures is inappropriate.
5. Chapter: "Unstimulated HGN-liposome effectively sequester bioactive compounds in vivo" shows the efficiency of substance storage very well. However, it does not answer the question of how the nanoparticles effectively goes to the brain and how they can get out of there, which should be the main question in this work.
6. Transport of nanoparticles to the brain is discussed in this paper, but the conclusion is that they are unlikely to penetrate the BBB. The authors do not present a convincing solution that makes sense. The proposed intravenous injection will lead the drug to the BBB, but the authors themselves admit that the drug may be stultified before the BBB to overcome it on its own. For what purpose, then, are nanoparticles needed? The drug will be distributed throughout the body, while a sophisticated solution should ensure its high concentration in the brain where it is needed. Nanoparticles will be eliminated quite fast, so the idea that they will be waiting for the weeks for "signal" to release is unlikely.
- "However, liposomes may gain access to the brain as a result of the seizure activity itself (Friedman, 2011)" If the seizures are to increase the permeability of the BBB to HGN, it means that their appropriate concentration in the circulatory system must be maintained all the time. If the seizures trigger the penetration of the BBB, it will be a long time from triggering the seizures to stopping them, so there is no preventive effect.
- "or BBB function may be momentarily disrupted by the ultrasound allowing liposome penetration (Mesiwala et al., 2002)." If so, the US will also release muscimol from the HGN into the peripheral blood instead of the brain. In such a system, it is much easier to administer muscimol intravenously alone because the HGN will be excreted after a few hours anyway. The authors ignored the rapid process of HGN elimination from the body.
7. The authors ignore the effect on the body of the nanoparticles themselves. They don't say how they are eliminated, if they were transported to the brain, how would they be removed? Has the body itself been tested for nanoparticles? Is cumulation possible? (They have the possibility to accumulate in the liver and spleen). How long are they present? If the drug has to be administered intravenously, what are its properties? How long does it stay in the body, how long is it active? How much drug should be administered peripherally to achieve the desired result? How should the effective dose be recalculated? Does the drug need to be administered daily? Should the medication be replaced after an attack? If the trigger is laser light or ultrasound, how would the drug be released in the patient?
There is no answer to these questions.
Other studies show that Ag-nanoparticles are present in the body for a short time - after 24 hours, they are undetectable. Researchers suggest using nanoparticles instead of conventional day-to-day treatment, while the nanoparticles seem to have to be used in the same way. The need to administer them iv reduces the chances of their widespread use, especially when considering central administration.
see:
Bailly, AL., Correard, F., Popov, A. et al. In vivo evaluation of safety, biodistribution and pharmacokinetics of laser-synthesized gold nanoparticles. Sci Rep 9, 12890 (2019). https://doi.org/10.1038/s41598-019-48748-3
8. In the chapter "Animals", the authors do not describe crucial information: the species, age, diet, weight, and a number of animals. The division into groups, the number of animals is completely unclear at work. It needs to be completed.
9. In an "in vivo study", an experiment was conducted where the group was n = 3, n = 3, n = 4, n = 8. This number of animals in the group is too small to perform appropriate statistical significance. What statistical test was used?
10. The chapter "Statistical analysis" is missing - it is not known how the authors made the calculations. In the "Results" section, the values are given: "(F(1,31) = 63.49,
p < 0.001, F(1,31) = 177.24, p < 0.001, F(1,31) = 29.07, p < 0.001 respectively). As expected, there was no difference in the effect of PTZ between rats allocated to receive one or other
of treatments (0-1Hz, F (3,31) = 1.97, p = 1.39, 1-3Hz F (3,31) = 1.94, p = 0.144, , 3-5 Hz F(3,31) = 0.401, p = 0.753)" without explaining what tests was used, what groups of animals were they, how many animals were there, etc.
11. How was the ultrasound signal used for drug triggering? Near the animal? Or rather, it was a local effect within the head, or was the entire animal exposed to ultrasound? The question is whether muscimol was released in some brain region or throughout the body from where it was transported to the brain (and other body tissues).
12. In "In vivo seizure model" authors wrote: "ultrasound was delivered in bursts of 30 sec at 30% duty cycle at 1 MHz." while in another place, they analyze different frequencies: "0-1 Hz, 1-3 Hz, and 3-5 Hz". The authors give frequencies in MHz, Hz, without specifying whether it is for stimulation or EEG reading. (see Figure 6 and in the text).
13."In the hippocampal slice model system, seizures were induced by perfusion with low Mg2+ /high K+ ACSF" Please provide the concentrations in relation to the standard ACSF. Elsewhere, the authors provide ACSF Mg-free solution. Is it the same?
14. There is no chapter "conclusions" where the most important conclusions and further questions would be collected.
Overall, the work has great potential, and sophisticated tools were used, new techniques and drug forms were presented. Unfortunately, the work is written chaotically; significant chapters are missing or incomplete, it is difficult to understand the pattern of the experiment and the subsequent stages step by step.
The work requires a deep revision.
Author Response
Response to Reviewers
We thank the reviewers for their careful reading of the manuscript and helpful comments. We have undertaken major revisions to address all the points of the referees, and we believe the paper is substantially improved as a result of their input.
Reviewer 3
Reviewer Comment
The authors conducted sophisticated experiments using advanced solutions.
However, many issues raise doubts.
minor issues:
- The authors did not use line numbering, which makes it difficult to explain by the reviewer where the errors are.
Author Response
We apologise for the lack of line numbers and have now added them.
Reviewer Comment
- At the beginning, the authors explain the abbreviations (PTZ, HGN, ACS). Why only these? There are many others (SLE, LRD, GLM, KA, ACSF, etc.). Maybe it is not necessary in this place, only in text.
Author Response
We have removed the list of abbreviations and defined abbreviations where they occur in the text.
Reviewer Comment
- Figure descriptions should be on the same page as figures.
Author Response
The document is now formatted so that figure descriptions appear on the same page as figures.
Reviewer Comment
- Figure 2 A, B, C - no scale, units, no explanations. The drawings are small, while in figure 3 the graphics are too large. You can make them smaller and put them side by side — the same style for all figures.
Author Response
We have further clarified that the vertical and horizontal lines on the traces are scale bars and have made them more obvious by making the lines thicker and coloring them blue. We collected the explanations of the scale bars and units at the end of the caption for all the panels. We have made Figure 3 smaller and the same style.
Reviewer Comment
- "In hippocampal brain slices repetitive electrical stimulation of Schaffer collaterals in area CA1 in the presence of high K +, Mg2 + -free conditions produced SLEs (6 slices from 6 animals, Figure. 2A)." "In wedges of entorhinal cortex, Mg2 + -free solution induced spontaneous SLEs (43 slices from 32 animals, Figure. 2B)." What do the graphs in Figures 2A, 2b show if "6 slices from 6 animals", "43 slices from 32 animals" are described?
Author Response
We apologize for any confusion. We have rewritten this section to make it clearer (lines 281-291), as copied below:
“Seizure-like activity was reliably induced in all three preparations, as illustrated in Figure 2. In hippocampal brain slices repetitive electrical stimulation of Schaffer collaterals in area CA1 in the presence of high K+, Mg2+-free conditions produced SLEs reliably in 6 slices from 6 animals. An example trace from these slices showing induction of SLEs by electrical stimulation in hippocampal area CA is shown in Figure 2A.
In wedges of entorhinal cortex, K+, Mg2+-free solution induced spontaneous SLEs. These began 40 to 120 min after switching to high K+, Mg2+-free solution. An example trace of a spontaneous SLE from this set of wedges is shown in Figure 2B. In 20 cases out of 43 wedges from 32 animals (i.e. in 47% of wedges) the SLEs spontaneously transitioned to faster, shorter and more continuous LRDs. An example of this transition is shown in Figure 2C. “
Reviewer Comment
- Figure 3A no description of the scale
Author Response
We have added a description of the scale to the figure caption.
Reviewer Comment
- figure 6. It is not explained what "time point" means. No statistical significance was noted. Were the groups differ from each other? The term "Muscisomes" is not explained in the text or in the figure.
Author Response
An explanation of the time points has been added to the figure caption. The term “Muscisomes” meant muscimol-filled HGN-liposomes, and has been changed to “PTZ + Musc + HG – Lipo + US” etc. in the figure.
Reviewer Comment
- If at the beginning of the "Results" section the authors refer to Figure 6, it should be shown here, not 5 pages later.
Author Response
We have moved the text relating to Figure 6 to later in the results section, close to the figure.
Reviewer Comment
- "Amelioration of seizure activity by muscimol release from liposomes in vitro" If this is a chapter name, it should be italic + period.
Author Response
Italics and period has been added.
major issues:
Reviewer Comment
- The critical question that arises is why the authors decided to use muscimol? We only learn from the work that it is "(GABA) agonist, muscimol". Information about this substance must be added. Why this substance was chosen, mechanism of action, duration of action, elimination. How was the dose selected? Has the accuracy and repetitiveness of the muscimol release? How many times it can be released?
Author Response
We have added more information about muscimol and the justification for choosing it, its mechanism, duration of action and elimination to the Introduction in two new paragraphs (lines 62-86), copied below”
“We used muscimol (3-hydroxy-5-aminomethylisoxazole) in the present study because it is a potent and selective GABAAreceptor agonist used extensively in electrophysiological studies of GABAergic inhibitory neurotransmission. Muscimol potently and reversibly inhibits neuronal activity, and thus has been considered to have the potential to suppress an epileptic seizure [7]. The metabolism of muscimol in both the brain and periphery is largely through the removal of an amino group by transamination [8]. In the mouse, about 1/3 is excreted as muscimol, 1/3 as a cationic conjugate, and 1/3 as an oxidation product [9]. The rapid clearance of muscimol in the periphery, and its slow passage across the blood-brain barrier (BBB), mean that high doses are needed when given intravenously, causing adverse effects making it unsuitable for systemic use in the treatment of epilepsy [10, 11]. “
“In contrast to systemic administration, when delivered transmeningeally in experimental animals, muscimol has antiepileptic effects without the adverse effects associated with systemic delivery [12-15]. Direct injection of muscimol into the brain is orders of magnitude more effective than intravenous injection. For example, nanomomolar concentrations injected into brain produce similar effects to micromolar concentrations injected intravenously [16, 17]. When injected locally into the brain in low μg quantities muscimol produced no sedation or other central side-effects [18, 19]. Similarly, studies of convection enhanced delivery of muscimol into the brain of non-human primates and patients with drug-resistant epilepsy, and other disorders, have shown that it is safe with no adverse effects [20-22]. Thus muscimol is a potential anti-epileptic treatment with few side effects provided it can be delivered directly to the brain. Muscimol is therefore a suitable candidate for proof of concept of the HGN-liposome delivery system.”
We have added text regarding dose, accuracy and repetitiveness of release from liposomes to the Discussion (Lines 505-522), copied below:
“In the present study we used laser stimulation to cause release of muscimol from HGN-liposomes in brain slices. Several previous studies have established that substances can be released from HGN-liposome nanostructures on a rapid timescale by laser stimulation in non-biological assays [27, 33-35]. These studies demonstrated the feasibility of drug delivery on a rapid timescale using laser stimulation. We have also previously shown that release and dosage can be controlled by varying the number and intensity of femtosecond pulses of light [27], and further, that on-demand release of different neurochemicals and drugs from HGN-liposome in live brain tissue has rapid, repeatable, and reliable physiological effects [28]. However, at present, laser-stimulated release from HGN-liposomes is not suitable for in vivo use because light does not penetrate far through the skull or brain parenchyma, and miniature femtosecond pulsed lasers are not available for chronic implantation. On the other hand, focused US can be transmitted through the skull and brain tissue. Recent work has demonstrated the feasibility of US-stimulated release of drugs from liposomes in vivo [2]. We have also shown that in vitro, US can evoke multiple release events of constant amount over 25 individual applications [6].The present study extends this work by showing that transcranial US stimulation both ex vivo and in vivo can cause sufficient release of muscimol from HGN-liposomes to arrest ongoing seizure activity. “
Reviewer Comment
- The purpose of the study is not to test the effectiveness of the new compound but the new method of its administration. So it seems obvious that the best choice at this point would be proven drugs used in the treatment of seizures (diazepam, clobazam) or progabide - approved for either monotherapy or adjunctive use in the treatment of epilepsy. These medications are safe and effective. Muscimol is a potent, selective agonist for the GABAA receptors but displays sedative-hypnotic, depressant and hallucinogenic psychoactivity as well. Next, it can produce euphoria, dream-like (lucid) state of mind, out-of-body experiences and synesthesia, negative effects include mild to moderate nausea, stomach discomfort, increased salivation and muscle twitching or tremors. In large doses strong dissociation or delirium may be felt. Were risk of those side effects were examined?
Muscimol was in clinical trial phase I for epilepsy, but the trial was discontinued.
Lonser, Russell R.; Oldfield, Edward H.; Sato, Susumu; Rene' Smith, R. N.; Walbridge, Stuart; Heiss, John D. (2012-08-01). "174 Convection-Enhanced Delivery of Muscimol to the Epileptic FocusPreclinical and Clinical Research". Neurosurgery. 71 (2): E568. doi:10.1227/01.neu.0000417764.02569.dc. ISSN 0148-396X.
Author Response
We agree that muscimol is unsuitable given systemically. The literature is clear, including the clinical trial mentioned, that there is potential to use muscimol locally by direct intracranial delivery.
We included additional references to support these points. We aim to avoid the systemic side effects by packaging the drug inside liposomes, and then to produce potent effects by releasing the drug in small quantities locally in the brain. We have added text to the Introduction (lines 74-85) and Discussion (lines 486-502) expanding points, copied below:
(lines 74-85) “In contrast to systemic administration, when delivered transmeningeally in experimental animals, muscimol has antiepileptic effects without the adverse effects associated with systemic delivery [12-15]. Direct injection of muscimol into the brain is orders of magnitude more effective than intravenous injection. For example, nanomomolar concentrations injected into brain produce similar effects to micromolar concentrations injected intravenously [16, 17]. When injected locally into the brain in low μg quantities muscimol produced no sedation or other central side-effects [18, 19]. Similarly, studies of convection enhanced delivery of muscimol into the brain of non-human primates and patients with drug-resistant epilepsy, and other disorders, have shown that it is safe with no adverse effects [20-22]. Thus muscimol is a potential anti-epileptic treatment with few side effects provided it can be delivered directly to the brain. Muscimol is therefore a suitable candidate for proof of concept of the HGN-liposome delivery system.”
(lines 486-502) “We used muscimol in these proof-of-concept experiments, rather than approved medications, for several reasons. Firstly, we were aiming for immediate seizure suppression, on demand, and focally at the site of seizure generation. Muscimol is both potent and rapidly-acting, thus is suitable for this approach. As shown in the present paper, and earlier studies [12-15] muscimol is immediately effective in attenuating seizures when applied locally at the site of the seizures. In contrast, the approved drugs are optimized for systemic treatment with minimal side effects, require several dosing cycles to make them effective at stopping seizures, and do not act as quickly when locally applied. Thus, because they are not optimal for local application and generally effective given systemically, there is less value in using them in HGN-liposome delivery. Secondly, about one-third of people with epilepsy have seizures refractory to systemic pharmacotherapy with approved medications [32]. For these people in particular, new approaches are needed. Third, as noted by Gernert [32], targeted intracranial delivery, by providing higher drug concentrations in localised target regions and lower concentrations in other brain or peripheral areas, allows the use of drugs that are otherwise unsuitable for systemic administration because of their toxicity or poor uptake into the brain. Since intracranial delivery of muscimol, in small quantities, has been shown to be safe in previous studies [11-15, 18-20] we used it as a test of the delivery system.”
Reviewer Comment
- Effects of muscimol last for 5–10 hours because it is not metabolized but eliminated in urine. If you prepare "on-demand drug-delivery system" and it should be activated before (?) sizers, it should end its action rapidly as well. If not, how can we control its action? What if overdose?
A good example of a treatable drug is using remifentanil in anesthesia. Its action takes 2-5 minutes so this feature is very useful in regulating deepness of action.
We have added text addressing these points. Although we agree that orally ingested muscimol has long-lasting effects, our intracellular recordings of the timecourse of the membrane potential hyperpolarisation after release from HGN-liposomes show that the timecourse of the inhibitory effect of muscimol is extremely short, on the order of seconds (Results lines 31-323; Discussion, lines 505-511). The ability to control the amount of release by varying the intensity and duration of stimulation allows the dose to be titrated against the effect (Discussion, lines 517-520). With the very small quantity of muscimol released from HGN-liposomes with each stimulus, overdose is unlikely, and previous studies have shown that direct cerebral infusion of muscimol in low concentrations does not cause adverse effects (Introduction lines 79-85; Discussion lines 499-504). The text is too extensive to copy here.
Reviewer Comment
The authors described epilepsy as an ideal candidate for "drug-on-demand" treatment. An epilepsy attack is usually unexpected and unpredictable. How would the patient release the drug before a seizure? If seizures do occur, the patient is usually unconscious or unable to act. If the seizure occurrence were to be a triggering factor, then the drug would not prevent seizures from occurring but would only act symptomatically when seizures occur. The goal of the currently approved as appropriate treatment of epilepsy is to take medications to PREVENT the occurrence of seizures, as their presence increases the risk of microdamage in the brain and the severity of epilepsy in the future. Treatment that does not provide protection but only treats existing seizures is inappropriate.
Author Response
We agree that ideally, treatment should prevent seizures. We also agree that existing technology is not yet capable of predictingseizures with clinically useful reliability. However, the technology for seizure detection already exists. We have added references and text supporting this in the Discussion (lines 589-606) as copied below:
“Ideally, treatment for epilepsy should prevent seizures before they occur. However, in one third of patients existing treatments are not effective in preventing seizures, and treatment resistant epilepsy is associated with significant morbidity and mortality [48, 49]. Existing technology is not yet capable of predicting seizures with clinically useful reliability. However, the technology for seizure detection already exists. For example, Kim et al [50] conclude from a review of the literature that “… the state-of-the-art seizure detection system performance is sufficient to build a robust and reliable wearable device that could be used for daily seizure monitoring and classification.” What is needed, therefore, is a means to deliver the drug immediately on the first sign of a seizure. Here we aimed to demonstrate proof of principle that seizures can be arrested almost instantaneously provided HGN-liposomes are preloaded in the brain.”
Reviewer Comment
- Chapter: "Unstimulated HGN-liposome effectively sequester bioactive compounds in vivo" shows the efficiency of substance storage very well. However, it does not answer the question of how the nanoparticles effectively goes to the brain and how they can get out of there, which should be the main question in this work.
Author Response
We agree that this is an important question for future development of this technology and we have added consideration of this matter to the Discussion (lines 564-582) as copied below.
“Even if biocompatibility issues can be overcome, clinical application of muscimol-containing HGN-liposomes will not be feasible until future technological developments provide a practical means to infiltrate them into the brain parenchyma. In the present study, HGN-liposomes may have gained access to the brain as a result of the seizure activity itself [44], or by momentary disruption of the BBB by the US stimulation allowing liposome penetration [45]. The combination of systemic injection of HGN-liposomes with focal US stimulation might thus achieve high local concentration of muscimol in the brain or brain vasculature, with relatively low concentration in the periphery due to encapsulation within liposomes and dilution of the cerebrally released muscimol. However, procedures such as carotid or intracerebral injections, as used in the present study, are invasive neurosurgical procedures that might only be considered in intractable drug resistant epilepsy [32]. For routine use less invasive ways will be required to move HGN-liposomes across the BBB and into the brain. Trojan horse liposomes (THLs) may be a future possibility. THLs are pegylated liposomes with a receptor-specific monoclonal antibody targeted to receptors that can transport liposomes across the blood-brain barrier, such as the transferrin receptor [46, 47]. The antibody is conjugated to the surface of the THL and the transferrin receptor ferries the liposome across the BBB . Further work is needed to determine if HGN-liposomes can be transported intact from the blood into the brain by hijacking existing transport mechanisms.”
Reviewer Comment
- Transport of nanoparticles to the brain is discussed in this paper, but the conclusion is that they are unlikely to penetrate the BBB. The authors do not present a convincing solution that makes sense. The proposed intravenous injection will lead the drug to the BBB, but the authors themselves admit that the drug may be stultified before the BBB to overcome it on its own. For what purpose, then, are nanoparticles needed? The drug will be distributed throughout the body, while a sophisticated solution should ensure its high concentration in the brain where it is needed. Nanoparticles will be eliminated quite fast, so the idea that they will be waiting for the weeks for "signal" to release is unlikely.
- "However, liposomes may gain access to the brain as a result of the seizure activity itself (Friedman, 2011)" If the seizures are to increase the permeability of the BBB to HGN, it means that their appropriate concentration in the circulatory system must be maintained all the time. If the seizures trigger the penetration of the BBB, it will be a long time from triggering the seizures to stopping them, so there is no preventive effect.
- "or BBB function may be momentarily disrupted by the ultrasound allowing liposome penetration (Mesiwala et al., 2002)." If so, the US will also release muscimol from the HGN into the peripheral blood instead of the brain. In such a system, it is much easier to administer muscimol intravenously alone because the HGN will be excreted after a few hours anyway. The authors ignored the rapid process of HGN elimination from the body.
Response
We agree and we have rewritten that section of the Discussion taking the reviewer’s comments into account (lines 564-582). This is reproduced above as it also applied to the previous comment of the referee.
Reviewer Comment
- The authors ignore the effect on the body of the nanoparticles themselves. They don't say how they are eliminated, if they were transported to the brain, how would they be removed? Has the body itself been tested for nanoparticles? Is cumulation possible? (They have the possibility to accumulate in the liver and spleen). How long are they present? If the drug has to be administered intravenously, what are its properties? How long does it stay in the body, how long is it active? How much drug should be administered peripherally to achieve the desired result? How should the effective dose be recalculated? Does the drug need to be administered daily? Should the medication be replaced after an attack? If the trigger is laser light or ultrasound, how would the drug be released in the patient?
There is no answer to these questions.
Other studies show that Ag-nanoparticles are present in the body for a short time - after 24 hours, they are undetectable. Researchers suggest using nanoparticles instead of conventional day-to-day treatment, while the nanoparticles seem to have to be used in the same way. The need to administer them iv reduces the chances of their widespread use, especially when considering central administration.
see:
Bailly, AL., Correard, F., Popov, A. et al. In vivo evaluation of safety, biodistribution and pharmacokinetics of laser-synthesized gold nanoparticles. Sci Rep 9, 12890 (2019). https://doi.org/10.1038/s41598-019-48748-3
Author Response
- We are grateful for the supplied reference. We agree that the questions raised by the reviewer are important for eventual clinical application. We have added text to the Discussion considering those questions (lines 548-563).
“The biocompatibility, distribution and eventual fate of the liposome constituents and HGNs is less clear and possibly more of a concern than the distribution of muscimol. After intravenous injection liposomes in circulation might be sequestered in liver or spleen. They might also cause immune reactions peripherally or cellular changes after crossing the BBB. Some of the pitfalls have been reviewed recently [38]. Gold, the constituent of the HGN component, has been used medically in ionic form in treatment of human rheumatoid arthritis, and the literature concerning adverse reactions to ionic gold has been reviewed recently [39]. In rats, gold nanoparticles were found to be biocompatible and relatively innocuous after intravenous injection, but the highest accumulation was in spleen and least in brain [40]. Laser-synthesized gold nanoparticles are considered to be purer and safe for biomedical applications [41] without causing liver damage. However, studies of gold nanoparticle effects in mice have revealed increased rate of abortion and fetal abnormalities if given in the early pregnancy [42]. Reviews of this topic highlight the limited available evidence and need for more knowledge concerning the toxicity of HGN after injection [43]. Excretion of accumulated particles from the liver and spleen can take up to 3 to 4 months, indicating that further studies of the toxicity of HGNs is needed. “
Reviewer Comment
- In the chapter "Animals", the authors do not describe crucial information: the species, age, diet, weight, and a number of animals. The division into groups, the number of animals is completely unclear at work. It needs to be completed.
Author Response
We have added information about the number of animals, age and species to the Animals section (lines 99-115), as copied below:
“A total of 11 mice and 58 rats were used in the research. Animals were handled in accordance with protocols approved by the Okinawa Institute of Science and Technology Animal Care and Use Committee (ex vivo hippocampal seizure model) and the University of Otago Animal Ethics Committee (ex vivo entorhinal cortex seizure model, and in vivo seizure model). In the ex vivohippocampal experiments brain slices were obtained from n = 6 male 3 to 8 week old Swiss Webster mice. Mice were group housed with littermates on reversed light cycle, with free access to standard chow and water. An additional n = 5 male 3 to 5 week old Swiss Webster mice were used to test liposome ability to sequester contents in absence of stimulation. After injection, these mice were individually housed until perfused for histology. In the ex vivo entorhinal cortex experiments brain slices were obtained from 40 male and female 4-8 week old Wistar rats. In the in vivo experiments a total of n=18 male Wistar rats were used, group housed in standard open top cages under reverse light cycle and fed standard rat chow and water ad libitum. The targeted weight range was 250 to 300g. These were allocated to four groups unbiased by any animal-related factors (PTZ-only, n = 3; PTZ plus HGN-liposome, n = 3; US without liposomes, n = 4; and PTZ plus HGN-liposome plus US, n = 8).”
Reviewer Comment
- In an "in vivo study", an experiment was conducted where the group was n = 3, n = 3, n = 4, n = 8. This number of animals in the group is too small to perform appropriate statistical significance. What statistical test was used?
Author Response
We have explained the analysis in a new section describing our use of general linear mixed model (GLMM) analyses further dissected using Fisher’s least significant difference post-hoc analyses. In the in vivo experiments GLMM analysis was used due to the use of multiple control groups of smaller size.
(lines 271-277). “For statistical analysis of group differences in the ex vivo wedge experiments we used one-way analysis of variance (ANOVA) to test for overall group differences and Tukey’s multiple comparisons post-hoc test for contrasts. For statistical analysis of group differences in the in vivo experiments we used general linear model mixed model (GLMM) [30, 31] analysis of data further dissected by Fisher’s least significant difference post-hoc analyses. In the in vivo experiments GLMM analysis was used due to the use of multiple control groups of smaller size. “
Reviewer Comment
- The chapter "Statistical analysis" is missing - it is not known how the authors made the calculations. In the "Results" section, the values are given: "(F(1,31) = 63.49,
p < 0.001, F(1,31) = 177.24, p < 0.001, F(1,31) = 29.07, p < 0.001 respectively). As expected, there was no difference in the effect of PTZ between rats allocated to receive one or other
of treatments (0-1Hz, F (3,31) = 1.97, p = 1.39, 1-3Hz F (3,31) = 1.94, p = 0.144, , 3-5 Hz F(3,31) = 0.401, p = 0.753)" without explaining what tests was used, what groups of animals were they, how many animals were there, etc.
Author Response
To improve clarity we have added a section “Statistical analysis” (271-277) noted above, and revised the text in the Results section (lines 396-412) and in the legend of Figure 6 to explain what tests were used, the groups, and group sizes.
Reviewer Comment
- How was the ultrasound signal used for drug triggering? Near the animal? Or rather, it was a local effect within the head, or was the entire animal exposed to ultrasound? The question is whether muscimol was released in some brain region or throughout the body from where it was transported to the brain (and other body tissues).
Author Response
We appreciate that this information may have been hard to find. We have added details to the Methods section (lines 233-238) explaining that liposomes were stimulated using an ultrasound transducer coupled to an acoustic collimator and positioned on the dura above the cortex as shown in Fig 1.
Reviewer Comment
- In "In vivo seizure model" authors wrote: "ultrasound was delivered in bursts of 30 sec at 30% duty cycle at 1 MHz." while in another place, they analyze different frequencies: "0-1 Hz, 1-3 Hz, and 3-5 Hz". The authors give frequencies in MHz, Hz, without specifying whether it is for stimulation or EEG reading. (see Figure 6 and in the text).
Author response
We have now specified stimulation or EEG reading in the text and figures.
Reviewer Comment
13."In the hippocampal slice model system, seizures were induced by perfusion with low Mg2+ /high K+ ACSF" Please provide the concentrations in relation to the standard ACSF. Elsewhere, the authors provide ACSF Mg-free solution. Is it the same?
Author Response
This information is now provided in the Methods section as follows:
The composition of low Mg2+ /high K+ ACSF was the same except for (mM) 5.0 K+ and 0.5 Mg2+ (lines 171-172). For Mg2+-free ACSF, the MgCl2 was omitted (line 216)
Reviewer Comment
- There is no chapter "conclusions" where the most important conclusions and further questions would be collected.
Author Response
We have added a Conclusions section (lines 615-644).
“Conclusions
The present experiments demonstrated that the HGN-liposome formulation we have developed is able to encapsulate and contain muscimol, and release it in the brain in response to femtosecond laser or US stimulation. Release is rapid and immediate, causing fast and repeatable hyperpolarizations of neurons similar to physiological inhibitory postsynaptic potentials. In ex vivo seizure models, stimulation of muscimol loaded HGN-liposomes caused immediate suppression of spontaneous and electrically evoked seizure activity. Experiments also showed that ultrasound stimulation applied to the brain through the dura attenuated seizure activity induced by PTZ in rats given intravenous injection of muscimol loaded HGN-liposomes. We also showed that intracerebrally injected HGN-liposomes loaded with toxic concentrations of KA did not cause damage to surrounding tissue. Thus, we demonstrate the feasibility of precise temporal control over exposure of neurons to the drug, potentially enabling therapeutic effects without continuous exposure. Overall, these findings suggest that HGN-liposomes combined with ultrasound triggering have potential for the development of innovative treatment strategies for neurological disorders using on-demand release of pharmaceuticals.
The present study focused on epileptic seizures in particular because of the challenges of long-term treatment with systemic antiepileptic drugs, and the large number of patients with drug-resistant epilepsy. The ability to deliver high concentrations of drug to target areas on demand while keeping drug concentrations low at other sites and times may enable the use of drugs that are effective applied locally, but unsuitable for systemic use because of their effects on other systems. Muscimol is one example of such a drug, which has been found unsuitable for systemic application but potentially effective when applied locally. For such applications the development of technology to move HGN-liposomes across the BBB and anchor them with the brain parenchyma would be necessary. Further work is needed to determine the utility and safety of the technology in humans, particularly concerning the pharmacokinetics, pharmacodynamics, and toxicity of HGN-liposomes and their constituents. Technology for detection of seizures and application US stimulation in ambulatory patients will also be needed. If these problems can be solved the HGN-liposomes have the potential to be developed into a new treatment for responsive forms of epilepsy.”
Reviewer Comment
Overall, the work has great potential, and sophisticated tools were used, new techniques and drug forms were presented. Unfortunately, the work is written chaotically; significant chapters are missing or incomplete, it is difficult to understand the pattern of the experiment and the subsequent stages step by step.
The work requires a deep revision.
Author Response
We appreciate the reviewer’s comments regarding the potential of the work. We have extensively revised and rewritten the paper according to the reviewer’s suggestions and believe it is much improved as a result.
Round 2
Reviewer 2 Report
NA
Reviewer 3 Report
Dear Authors,
After a thorough revision, the work has the correct layout, it is more legible and understandable.
I am not a fan of this type of work - the authors have put a lot of effort into carrying out the research, however, the research is very unrealistic. In my research I want to see a real goal - here we have several separate, isolated results, but this method cannot be applied at the moment due to numerous limitations that cannot be overcome at the moment. Thus, the work, although interesting, remains in the sphere of hypothetical works.